# Polymerase delta-interacting protein 38 (PDIP38) modulates the stability and activity of the mitochondrial AAA+ protease CLPXP

Philip R. Strack[1], Erica J. Brodie[1,5], Hanmiao Zhan[1], Verena J. Schuenemann[2,6], Liz J. Valente[1,7], Tamanna Saiyed[1], Bradley R. Lowth[1], Lauren M. Angley[3], Matthew A. Perugini[1,3], Kornelius Zeth[2,4], Kaye N. Truscott [1,8✉] & David A. Dougan [1,8✉]

Over a decade ago Polymerase δ interacting protein of 38 kDa (PDIP38) was proposed to play a role in DNA repair. Since this time, both the physiological function and subcellular location of PDIP38 has remained ambiguous and our present understanding of PDIP38 function has been hampered by a lack of detailed biochemical and structural studies. Here we show, that human PDIP38 is directed to the mitochondrion in a membrane potential dependent manner, where it resides in the matrix compartment, together with its partner protein CLPX. Our structural analysis revealed that PDIP38 is composed of two conserved domains separated by an α/β linker region. The N-terminal (YccV-like) domain of PDIP38 forms an SH3-like β-barrel, which interacts specifically with CLPX, via the adaptor docking loop within the N-terminal Zinc binding domain of CLPX. In contrast, the C-terminal (DUF525) domain forms an immunoglobin-like β-sandwich fold, which contains a highly conserved putative substrate binding pocket. Importantly, PDIP38 modulates the substrate specificity of CLPX and protects CLPX from LONM-mediated degradation, which stabilises the cellular levels of CLPX. Collectively, our findings shed new light on the mechanism and function of mitochondrial PDIP38, demonstrating that PDIP38 is a bona fide adaptor protein for the mitochondrial protease, CLPXP.

[1] Department of Biochemistry and Genetics, La Trobe Institute for Molecular Science, La Trobe University, Melbourne, VIC 3086, Australia. [2] Department of Protein Evolution, Max-Planck-Institute for Developmental Biology, Tübingen 72076, Germany. [3] Department of Biochemistry and Molecular Biology, The University of Melbourne, Parkville, VIC 3010, Australia. [4] Department of Science and Environment, Roskilde University, DK-4000 Roskilde, Denmark. [5] Present address: CSL Behring, Broadmeadows, VIC 3047, Australia. [6] Present address: Institute of Evolutionary Medicine, University of Zurich, Zurich, Switzerland. [7] Present address: Department of Radiation Oncology, Stanford University School of Medicine, Stanford, CA 94305, USA. [8] These authors contributed equally: Kaye N. Truscott, David A. Dougan. ✉email: k.truscott@latrobe.edu.au; d.dougan@latrobe.edu.au

Mitochondria are essential organelles that play crucial roles in energy transduction, haem biosynthesis, fatty acid oxidation and cell signalling. Although the vast majority of mitochondrial proteins (~99%) are encoded on nuclear DNA, human mitochondria also contain a small genome (mtDNA) that encodes 13 polypeptides. This distribution of mitochondrial genes (over two genomes) creates a number of challenges for mitochondria, most notably the maintenance of protein homeostasis (proteostasis) within the organelle. Consistent with the importance of proteostasis within this organelle, genetic mutations in components of the mitochondrial proteostasis network (PN) have been linked to mitochondrial dysfunction, disease and aging[1,2].

A key aspect of mitochondrial proteostasis is the regulated removal of damaged or unwanted proteins. In humans, this process is performed by five different AAA+ (ATPases associated with a variety of cellular activities) proteases, three of which are located in the inner membrane (two forms of the *m*-AAA (*matrix*-AAA) protease and a single *i*-AAA (*intermembrane space*-AAA) protease[3]) and two of which are located in the matrix, LONM (encoded by a single gene, *LONP1*) and CLPXP (encoded by two genes, *CLPX* and *CLPP*). Although deletion of *LONP1* is embryonic lethal[4] and LONM is regarded as the principal AAA+ protease of the matrix compartment[5–8], genetic mutations in *CLPX* and *CLPP* are associated with severe phenotypes in mice and diseases in humans[9–11]. Consistently, mammalian CLPXP is reported to play a critical role in a variety of important functions, including mitoribosome assembly[12] and haem regulation[11,13,14]. Likewise, compounds that dysregulated mitochondrial CLPP[15] exhibit therapeutic potential against specific cancer cells[16–18]. Despite the emerging importance of this protease in human health and disease, only a handful of CLPXP substrates have been verified. Furthermore, given human CLPX exhibits a distinct substrate specificity relative to its bacterial homologues[15,19], our current understanding of this proteolytic machine is limited.

Like most AAA+ proteases[20–23], human CLPXP is composed of two components: a peptidase component, CLPP and a AAA+ unfoldase component, CLPX[24–26]. The peptidase component forms a barrel-shaped oligomer that is composed of two heptameric rings, stacked back-to-back. To prevent the indiscriminate turnover of proteins, the catalytic residues of CLPP are encapsulated within the barrel and access to the proteolytic chamber is limited to a narrow entry portal at either end of the cylinder. The unfoldase component CLPX, forms a hexameric ring that sits at one or both ends of CLPP acting as a gatekeeper to the proteolytic chamber. As such, CLPX is not only responsible for recognition of the substrate, but also for its ATP-dependent unfolding and translocation into the proteolytic chamber of CLPP[27]. In the case of *Escherichia coli* ClpX (*ec*ClpX), substrate recognition is mediated by various distinct loops and regions (i.e. pore loops and accessory domains), some of which are exclusively required for particular substrates[19,28,29]. In addition to the direct recognition of substrates, some AAA+ proteases also use specialised cofactors (commonly referred to as adaptor proteins) to alter substrate recognition and/or specificity[30–34]. Currently, however, no AAA+ adaptor proteins have been identified in mammalian mitochondria.

Previously, we identified mouse polymerase δ-interacting protein of 38 kDa (PDIP38, also known as POLDIP2) as a putative mitochondrial CLPX-interacting protein[15]. However, since its discovery in a yeast two-hybrid screen, as a p50 and PCNA-interacting protein[35], the subcellular distribution of PDIP38 and its physiological function have remained perplexing. Indeed, over the past 20 years, there have been several conflicting reports examining the subcellular location of tagged or endogenous PDIP38. To date, PDIP38 has been identified in the nucleus[36,37], the mitochondrion[15,38], the cytoplasm[39–41] and the plasma membrane[39,40], where it is proposed to mediate a variety of cellular functions, including cell proliferation[40], regulation of the extracellular matrix[42], oxidative signalling and cell migration[43], Tau aggregation[44], cancer[45] and DNA repair[37,39,41,46]. However, several of these cellular studies have incorporated EGFP-tagged versions of PDIP38 (which when attached to the N terminus of PDIP38 prevent its targeting to the mitochondrion) and very few of the putative interactions have been validated in vitro. As such, the physiological significance of some of these proposed functions remain uncertain. Nevertheless, consistent with our previous identification of PDIP38 as a CLPX-interacting protein, Martin and colleagues[47] recently proposed that PDIP38 controls the lipoylation of pyruvate dehydrogenase and α-ketoglutarate dehydrogenase complexes by inhibiting CLPXP. However, the details of its interaction with CLPX and how it regulates CLPXP activity are currently unknown.

Here we show that human PDIP38 is targeted to the mitochondrion via an N-terminal presequence and imported into the matrix compartment in a Δψ-dependent manner, where it colocalises with its partner proteins CLPX and CLPP. Critically, PDIP38 is neither degraded by CLPXP nor does it trigger dissociation of the CLPXP complex. Rather, PDIP38 modulates the specificity of CLPXP in vitro and alters the stability of CLPX both in vitro and in cells. Hence, we propose that mitochondrial PDIP38 is a specialised adaptor protein for the CLPXP protease. Consistent with this role, PDIP38 is composed of two domains, an N-terminal "YccV-like" domain, which docks to a specific adaptor binding loop within the zinc-binding domain (ZBD) of CLPX, and a C-terminal domain of unknown function (DUF525), which forms an immunoglobulin-like fold. Interestingly, although the binding repertoire of PDIP38 is currently unknown, based on our PDIP38 structure and its similarity to Fbxo3 (the substrate recognition component of a Skp Cullin F-box (SCF) E3 ligase), we propose that PDIP38 recognises a ligand via a conserved pocket. Importantly, mutation of the adaptor-docking loop in human CLPX specifically inhibited PDIP38 docking, without effecting peptide (substrate) binding. Taken together, we propose that mitochondrial PDIP38 represents the first adaptor protein for the AAA+ protease, CLPXP.

## Results

**Human PDIP38 is targeted to the mitochondrial matrix, where it associates with CLPX.** Although PDIP38 was originally described as a nuclear protein, it has been reported to localise to several subcellular compartments, including the mitochondrion, the cytosol and even the plasma membrane. As a result of the broad and uncertain subcellular distribution of PDIP38, its physiological function currently remains unclear. Therefore, in order to resolve this ambiguity, we examined the in vitro import of radiolabelled human preprotein (pre-PDIP38) into isolated mitochondria and performed mitochondrial fractionation experiments (Fig. 1). Consistent with previous studies from ours and other groups[15,38,47], these in vitro import data clearly demonstrate that the PDIP38 preprotein (pre-PDIP38) is imported into isolated mitochondria in a membrane potential-dependent manner (Fig. 1a, compare lanes 11 and 12). Importantly, the processed, mature (mt) form of the protein (mt-PDIP38) was protected from cleavage by proteinase K (Prot. K), demonstrating that mt-PDIP38 (herein referred to as PDIP38) was sequestered inside mitochondria. As expected for a matrix localised protein, PDIP38 was protected from digestion by Prot. K in both intact mitochondria (Fig. 1b, lanes 1–4) and mitoplast (Fig. 1b, lanes 5–8), similar to the known matrix protein CLPP

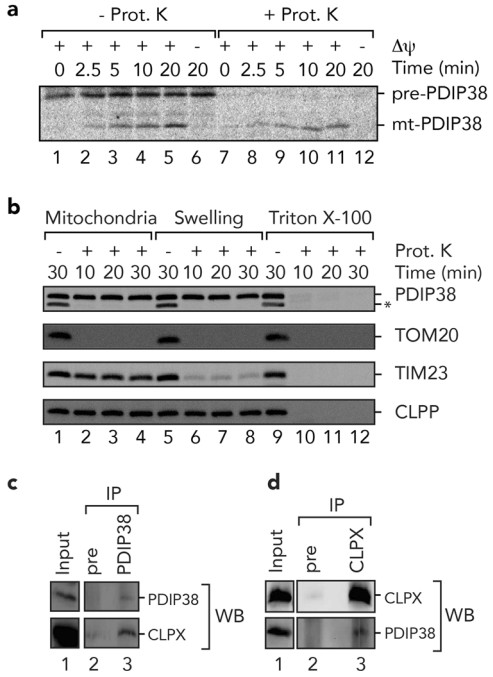

**Fig. 1 Human PDIP38 is imported into mitochondria where it interacts with CLPX. a** Import of [$^{35}$S]-labelled PDIP38 preprotein into mitochondria isolated from HeLa cells, in the presence or absence of a membrane potential (Δψ) as indicated, treated with (lanes 7–12) or without (lanes 1–6) proteinase K (Prot. K). Samples were separated by 15% Tris-glycine SDS-PAGE and analysed by digital autoradiography. Full gel is shown in Supplementary Fig. 13. **b** Mitochondria were incubated, either in an osmotic buffer (lanes 1–4), isotonic buffer (Swelling) to rupture the outer membrane (lanes 5–8) or buffer containing Triton X-100 (lanes 9–12), in the absence (lanes 1, 5 and 9) or presence (lanes 2–4, 6–8 and 10–12) of Prot. K for the indicated time. Samples were separated by 16.5% Tricine-buffered SDS-PAGE and subjected to immunoblotting with the appropriate antisera to visualise endogenous proteins. *Non-specific cross-reactive protein recognised by PDIP38 antisera. Full immunoblots are shown in Supplementary Fig. 14. **c** Specific immunoprecipitation (IP) of endogenous PDIP38 from detergent-solubilised mitochondria using anti-PDIP38 polyclonal antibodies showing co-immunoprecipitation of endogenous CLPX. The input represents 50% of total mitochondrial lysate subjected to IP. Pre preimmune serum. Full immunoblot strips are shown in Supplementary Fig. 13. **d** Specific immunoprecipitation of endogenous CLPX from detergent-solubilised mitochondria using anti-CLPX polyclonal antibodies showing co-immunoprecipitation of endogenous PDIP38. The input represents 50% of total mitochondrial lysate subjected to IP. Full immunoblot strips are shown in Supplementary Fig. 13.

(Fig. 1b, bottom panel). As a control, the outer membrane protein (TOM20) was digested by Prot. K under all conditions, while the inner membrane protein (TIM23) was completely protected in intact mitochondria, but sensitive in mitoplast (Fig. 1b, lanes 5–8). Next, having confirmed that human PDIP38 is a mitochondrial matrix protein, we examined the interaction between CLPX and PDIP38 in human mitochondria by co-immunoprecipitation (co-IP). Initially, we monitored the interaction of endogenous CLPX (with endogenous PDIP38), using a PDIP38-specific antisera. Consistent with a specific interaction between PDIP38 and CLPX, CLPX was only recovered in the presence of protein A sepharose (PAS)-immobilised anti-PDIP38 antibodies (Fig. 1c, lane 3, lower panel) and not in the presence of PAS-immobilised preimmune antibodies (Fig. 1c, lane 2, lower panel). Next, to confirm this interaction, we performed the reverse co-IP, in which antibodies from anti-CLPX antisera were

immobilised to PAS. Consistent with the specific IP of CLPX with anti-PDIP38 antibodies, the IP of CLPX using anti-CLPX antibodies also resulted in the specific co-IP of PDIP38 (Fig. 1d, lane 3, lower panel). Next, to confirm the interaction, we immobilised a His$_{10}$-tagged version of recombinant PDIP38 ($_{H10}$PDIP38) to Ni-NTA agarose and performed a pull-down using detergent-solubilised HeLa mitochondria. Consistent with the co-IPs above, CLPX was only recovered from the column containing immobilised PDIP38 (Supplementary Fig. 1, lane 6) and not from the control columns (Supplementary Fig. 1, lanes 5 and 7). Next, to ensure the interaction that we observed in cells (between PDIP38 and CLPX) was direct, we purified both components and examined the interaction in vitro. Specifically, we generated recombinant GST-PDIP38 fusion protein in which PDIP38 was fused to the C terminus of GST. Following expression of GST-PDIP38, a soluble lysate (bearing overexpressed GST-PDIP38) was applied to Ni-NTA agarose beads (either lacking or containing immobilised His$_{10}$-tagged CLPX). Following incubation of GST-PDIP38 with the beads and subsequent wash steps, H$_{10}$CLPX together with the bound proteins was eluted from the column with imidazole (Fig. 2a, lanes 4 and 7). Consistent with our identification of PDIP38 as a CLPX-interacting protein (Fig. 1), GST-PDIP38 was only recovered in the presence of immobilised H$_{10}$CLPX (Fig. 2a, lanes 7) and not in the absence of an immobilised protein (Fig. 2a, lanes 4).

**PDIP38 is neither a substrate of CLPXP nor does it trigger dissociation of the CLPXP complex.** Next, we examined the consequence of PDIP38 docking to CLPX, to determine if PDIP38 is a substrate of the CLPXP protease or a regulator that triggers dissociation of the CLPXP complex? To determine if PDIP38 is a substrate of CLPXP, we monitored the stability of purified recombinant mature PDIP38 in vitro, in the presence of active CLPXP (Fig. 2b). Given that substrate recognition by many Clp proteases is generally mediated by degrons located at either the N or C termini[48], we generated an untagged version of PDIP38, using the Ub-fusion system[49]. As a control, to ensure that human CLPXP was active, we monitored the turnover of casein, a model unfolded protein and well-characterised CLPXP substrate[15,26]. Significantly, in contrast to the rapid CLPXP-mediated turnover of fluorescein isothiocyanate (FITC)-labelled casein (FITC-casein) (Fig. 2b, middle panel), the levels of untagged PDIP38 remained unchanged throughout the time course of the experiment (Fig. 2b, upper panel). These data clearly demonstrate that mature PDIP38 is not a substrate of the CLPXP protease; however, it remained unclear if the lack of PDIP38 turnover was due to dissociation of the CLPXP complex (triggered by PDIP38 docking to CLPX). To examine this possibility, we monitored the CLPXP-mediated turnover of FITC-casein in the presence of PDIP38 (Fig. 2b, lower panel). Interestingly, the addition of PDIP38 produced distinct effects on the turnover of FITC-casein. Although the turnover of α$_{S2}$-casein was inhibited by PDIP38 (Fig. 2b, lower panel), in a concentration-dependent manner (Supplementary Fig. 2), the turnover of α$_{S1}$- and κ-casein was unaffected by the presence of PDIP38 (Fig. 2b, lower panel and Supplementary Fig. 2, black bars). Given that the CLPXP-mediated turnover of κ-casein was unchanged by PDIP38 addition, these data suggest that the CLPXP complex remains intact in the presence of PDIP38. To confirm that PDIP38 interacts not only with CLPX, but also with CLPXP, we examined complex formation using fluorescence detected analytical ultracentrifugation (FD-AUC). Specifically, we generated a GFP-PDIP38 fusion protein and examined its sedimentation (in the absence or presence of CLPX with or without the addition of CLPP). The use of GFP-PDIP38 allowed specific detection (by GFP fluorescence) of

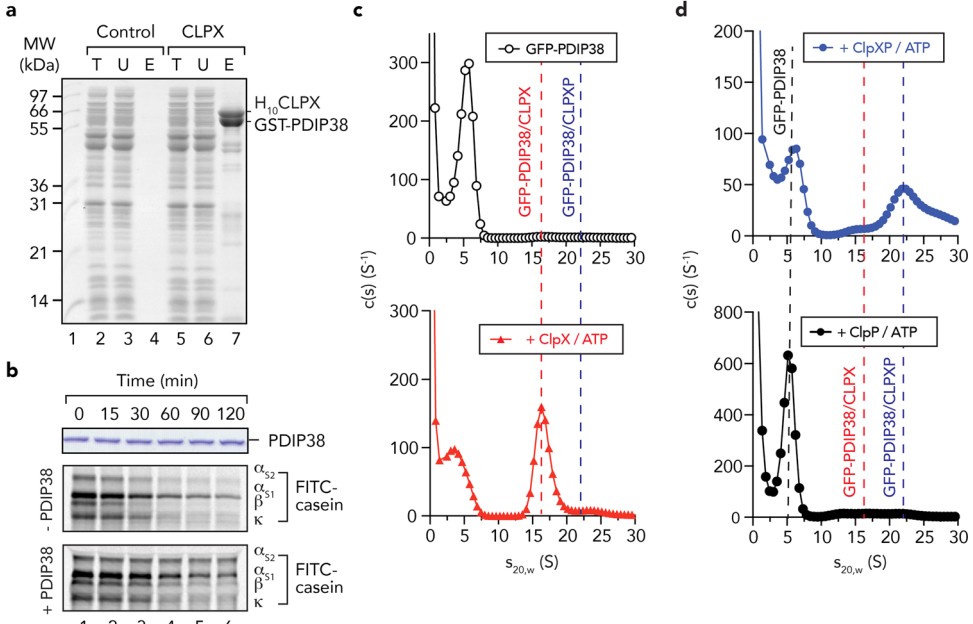

**Fig. 2 PDIP38 forms a complex with CLPX (and CLPXP), but is not a CLPXP substrate. a** Coomassie Brilliant Blue (CBB) stained 16.5% Tricine-buffered SDS-PAGE of PDIP38 pull-down from *E. coli* lysate containing recombinant GST-PDIP38, using beads either lacking (control) or containing immobilised $H_{10}$CLPX (CLPX). T total *E. coli* lysate with expressed GST-PDIP38, U unbound fraction, E eluted fraction. Lane 1, SeeBlue Plus MW protein standards. **b** In vitro degradation assay of PDIP38 (upper panel) or FITC-labelled casein by CLPXP protease in the absence or presence of 2.4 μM untagged PDIP38 (lower panels, as indicated). Samples were separated by 16.5% Tricine-buffered SDS-PAGE and analysed by fluorescence detection (FITC-casein) or CBB staining (PDIP38). Full gels are shown in Supplementary Fig. 15. **c, d** c(s) distribution profiles for GFP-PDIP38 in the presence of 5 mM ATP (open circles) with the addition of either CLPX$_{TRAP}$ (red triangles), CLPX$_{TRAP}$P (blue circles) or CLPP (black circles).

complexes that only contained PDIP38 and eliminated the detection of subcomplexes of CLPX and/or CLPXP that lacked PDIP38. To monitor the interaction of PDIP38 with CLPX and CLPXP complexes, all FD-AUC experiments were performed using the CLPX$_{TRAP}$ mutant[15] in the presence of 5 mM ATP. Initially, we examined the sedimentation of GFP-PDIP38 (Fig. 2c, open circles). As expected for a protein of ~65 kDa, GFP-PDIP38 sedimented in a single peak with an $s_{20,w}$ of ~5.8 S. Next, we examined the sedimentation of GFP-PDIP38 in the presence of CLPX. Consistent with the previously observed sedimentation of hexameric CLPX $s_{20,w}$ ~14 S[50], GFP-PDIP38 (in the presence of CLPX) sedimented with an $s_{20,w}$ of ~16.5 S (Fig. 2c, red triangles), indicating the GFP-PDIP38 interacts with hexameric CLPX. Finally, we examined the sedimentation of GFP-PDIP38 in the presence of both CLPX and CLPP. As expected, the sedimentation of GFP-PDIP38 shifted (from ~5.8 S) to a main peak of $s_{20,w}$ ~22 and a shoulder at $s_{20,w}$ ~28 (Fig. 2d, blue circles). These data are consistent with previous AUC analysis of CLPXP complexes[50], and the two peaks likely represent GFP-PDIP38 bound to single- and double-headed complexes of CLPXP. Nevertheless, to ensure that both peaks represented specific CLPXP complexes and were not due to a non-specific interaction of GFP-PDIP38 with CLPP, we also examined the sedimentation of GFP-PDIP38 in the presence of CLPP (Fig. 2d, black circles). Consistent with a specific interaction of PDIP38 with CLPX, the sedimentation of GFP-PDIP38 was unchanged in the presence of CLPP alone. Taken together, these data demonstrate that PDIP38 is neither a substrate of the CLPXP machine nor does it trigger dissociation of CLPX from CLPXP. Rather, PDIP38 specifically inhibits the turnover of one form of casein without affecting the turnover of another, and hence, similar to known AAA+ adaptor proteins[30,51–55], modulates the substrate selectivity of CLPX.

**PDIP38 docks to the "adaptor binding" loop within the ZBD of CLPX.** Next, we asked the question how does the CLPX-PDIP38 complex form? Based on the findings above, we speculated that PDIP38 could be an adaptor protein of human CLPX, and as such would likely bind to an accessory domain of CLPX, as is the case for several bacterial AAA+ adaptor proteins[28–30,52]. Interestingly, in contrast to bacterial ClpX homologues, human CLPX contains two accessory domains, an N-terminal C4-type zinc-finger domain (commonly referred to as a ZBD) and an additional domain that is inserted into the AAA module of CLPX, which is unique to eukaryotic CLPX homologues (termed the E-domain[56]). Therefore, to identify which CLPX domain (or domains) might be responsible for the interaction with PDIP38, we purified $H_{10}$-tagged versions of both the ZBD and the E-domain (Fig. 3a) of CLPX. These proteins were then immobilised to Ni-NTA agarose and a soluble lysate bearing GST-PDIP38 was applied to the appropriate columns (Fig. 3b, lane 2, 5 and 8). Consistent with the data above, in which PDIP38 was recovered in the presence of full-length CLPX, PDIP38 was specifically co-eluted from the column containing immobilised ZBD (Fig. 3b, lane 10) and not from the column containing immobilised E-domain (Fig. 3b, lane 7). Collectively, these data demonstrate that PDIP38 interacts specifically with the ZBD of CLPX. Next, we performed a series of pull-down experiments to examine the affinity of this interaction (Supplementary Fig. 3). From these data, we estimate that PDIP38 binds to the ZBD of human CLPX with a binding affinity ($K_d$ ~1.8 μM) that is similar to other AAA+ adaptor proteins[57–59]. We then examined the stoichiometry of the interaction between PDIP38 and the ZBD. To do so, we purified full-length PDIP38 (containing an N-terminal $H_{10}$-tag) and monitored complex formation by size-exclusion chromatography, using Superose 12. Consistent with its theoretical MW (38.3 kDa), PDIP38 eluted in a single peak (Fig. 3c, upper panel)

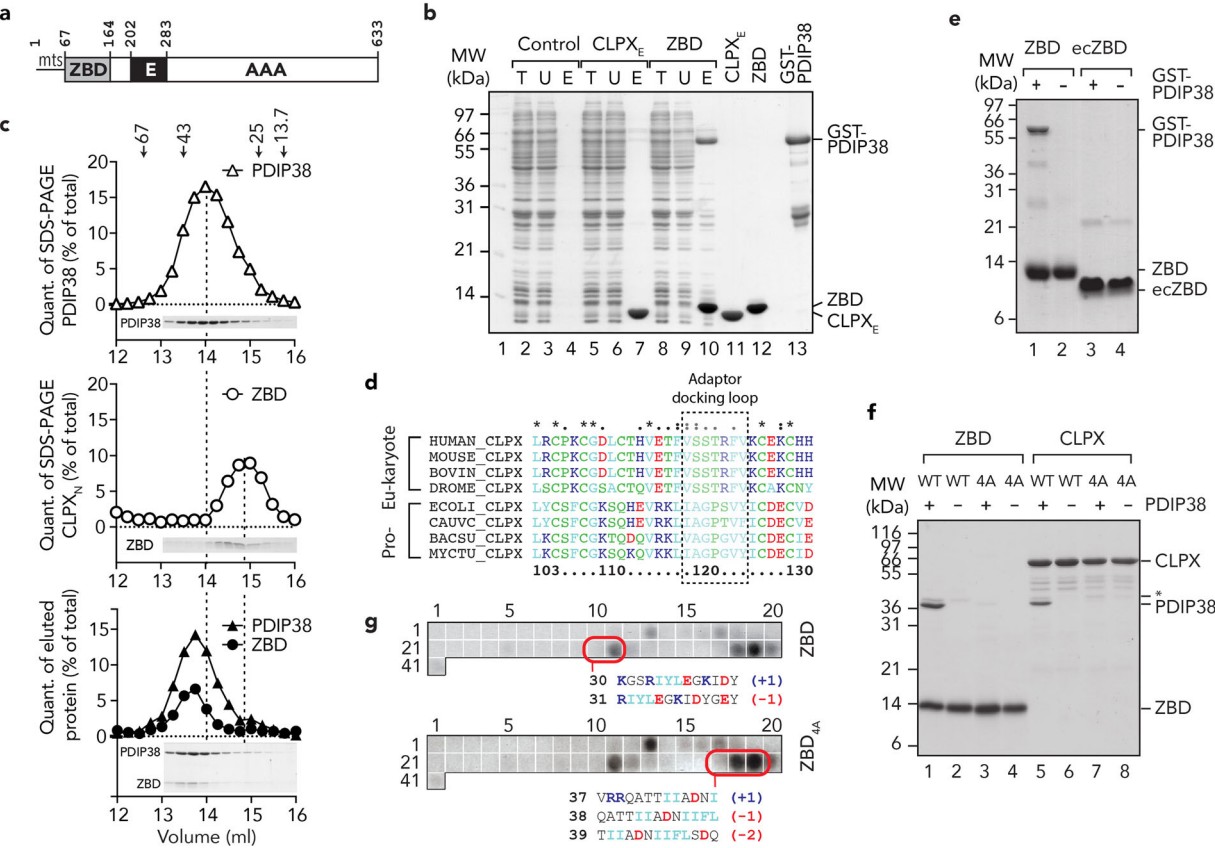

**Fig. 3 PDIP38 interacts with the adaptor-docking loop of human CLPX$_{ZBD}$. a** Schematic representation of human CLPX domain structure. mts mitochondrial targeting sequence, ZBD zinc-binding domain, E eukaryotic domain, AAA ATPase associated with a variety of cellular activities domain. **b** Coomassie-stained 16.5% Tricine-buffered SDS-PAGE of PDIP38 pull-down from *E. coli* lysate expressing recombinant GST-PDIP38 using bead immobilised H$_{10}$CLPX E-domain (CLPX$_E$) and H$_{10}$CLPX zinc-binding domain (ZBD) relative to beads only (control). T total lysate expressing GST-PDIP38, U unbound fraction, E eluted fraction. For comparison purified recombinant CLPX$_E$ (lane 11), ZBD (lane 12) and GST-PDIP38 (lane 13) are shown. **c** Complex formation of PDIP38 and ZBD was monitored by size-exclusion chromatography (SEC) using a Superose 12 column. Elution profiles of PDIP38 (upper panel), ZBD (middle panel) or PDIP38 in the presence of ZBD (bottom panel) were determined from quantification of protein bands (inset). Arrows indicate the peak elution volume of albumin (67 kDa), ovalbumin (43 kDa), chymotrypsin A (25 kDa) and ribonuclease (13.7 kDa). **d** Amino acid sequence alignment of eukaryotic and prokaryotic CLPX homologues showing part of the ZBD. The prokaryotic ClpX *adaptor-docking loop* is highlighted in the boxed section. ClpX sequences are *Homo sapiens* (O76031); *Mus musculus* (Q9JHS4); *Bos taurus* (F1N155); *Drosophila melanogaster* (Q9VDS7), *Escherichia coli* (P0A6H1), *Caulobacter crescentus* (P0CAU2), *Bacillus subtilis* (P50866) and *Mycobacterium tuberculosis* (P9WPB9). **e** In vitro pull-down, in which purified human (lanes 1 and 2) or *E. coli* (lanes 3 and 4) ZBD was immobilised to Ni-NTA agarose beads and incubated with (lanes 1 and 3) or without (lanes 2 and 4) an *E. coli* lysate containing expressed recombinant GST-PDIP38. Eluted fractions are shown with samples analysed by CBB staining. **f** In vitro pull-down, in which purified wild-type (WT, lanes 1 and 2) or mutant (4A, lanes 3 and 4) ZBD and wild-type (WT, lanes 5 and 6) or mutant (4A, lanes 7 and 8) human H$_{10}$CLPX was immobilised to Ni-NTA agarose beads and incubated with (lanes 1, 3, 5 and 7) or without (lanes 2, 4, 6 and 8) an *E. coli* lysate expressing untagged PDIP38. Eluted proteins were separated by SDS-PAGE and visualised by CBB staining. **g** Binding of wild type (ZBD) and mutant (ZBD$_{4A}$) to overlapping 13-mer peptides representing mature mtSSB. Peptide sequences for numbered spots are shown in Supplementary Fig. 4. Highlighted spots illustrate that the ZBD of human CLPX favours binding to peptides with a hydrophobic core and a net negative charge.

with an estimated molecular weight of ~45 kDa. In contrast, the ZBD of CLPX (Fig. 3c, middle panel) eluted at ~15 ml, (estimated MW of ~29 kDa), which is approximately double its theoretical MW (12.1 kDa) and hence similar to the other ClpX homologues, likely forms a homodimer (ZBD$_2$). Consistent with the pull-down (Fig. 3b and Supplementary Fig. 3), PDIP38 formed a stable complex with CLPX$_{ZBD}$, which is based on the elution volume of the complex (~13.6 ml, equivalent to ~60 kDa) is likely a heterodimeric complex composed of PDIP38 bound to ZBD$_2$ (theoretical MW = 62.5 kDa) (Fig. 3c, lower panel). Next, to gain a better understanding of the specificity of this interaction (between PDIP38 and CLPX), we compared the ZBD of several ClpX homologues, from both pro- and eukaryotic species (Fig. 3d), focusing on a particular region within the domain that is important for adaptor docking[28,60,61]. Interestingly, despite broad

conservation of the entire domain, there was considerable sequence divergence across the putative adaptor-docking region (Fig. 3d). Therefore, we hypothesised that this region may have co-evolved with a new adaptor protein (i.e. PDIP38). To test this idea, we compared the ability of the human CLPX ZBD and *ec*ClpX ZBD (*ec*ZBD) to interact with human PDIP38 (Fig. 3e). As predicted, human PDIP38 was exclusively recognised by human ZBD (Fig. 3e, lane 1) as GST-PDIP38 was not recovered in the presence of *ec*ZBD (Fig. 3e, lane 3). Next, to determine which part of the adaptor-docking region was required for interaction with PDIP38, we generated a specific mutation within this region, in which residues 120–123 (SSTR) of human CLPX were replaced with AAAA in either full-length CLPX (here referred to as CLPX$_{4A}$) or the ZBD of CLPX (here referred to as ZBD$_{4A}$). Consistent with the lack of binding of PDIP38 to *ec*ZBD,

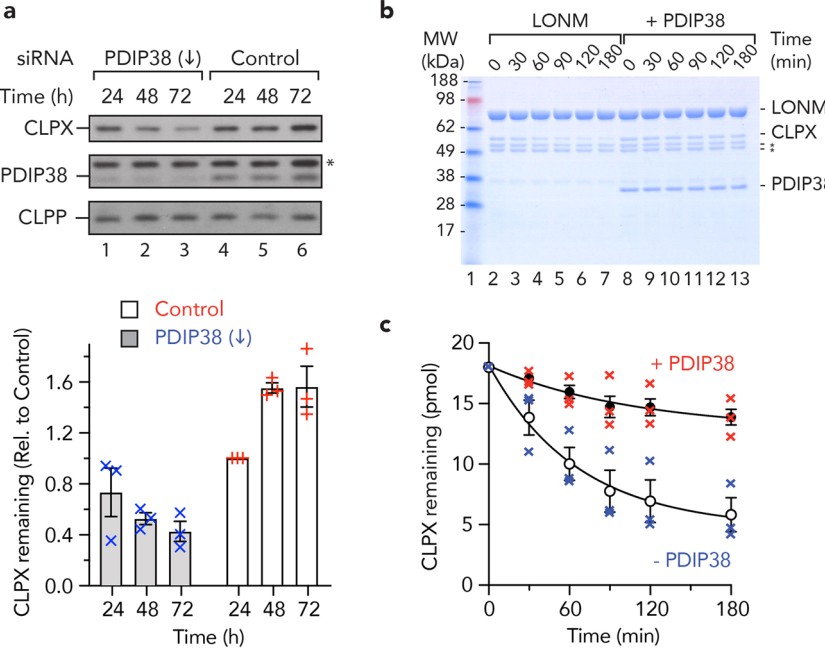

**Fig. 4 PDIP38 stabilises CLPX protecting it from LONM-mediated degradation. a** Representative immunoblots showing the steady-state levels of CLPX (upper gel strip), PDIP (middle gel strip) and CLPP (lower gel strip) in PDIP38-depleted HeLa cells (lanes 1–3) relative to control HeLa cells (lanes 4–6). Samples were collected at the indicated times post transfection of either Silencer Select siRNA (22994) targeting PDIP38 or a negative control siRNA (control). Proteins were separated by 15% Tris-glycine SDS-PAGE. *Non-specific cross-reactive protein in PDIP38 antisera. Full immunoblots are shown in Supplementary Fig. 16. The lower panel shows the quantitation of CLPX levels from three independent experiments, in PDIP38-depleted HeLa cells (grey bars, each blue cross represents an individual experiment) in comparison a negative control siRNA (white bars, each red plus represents an individual experiment). Initial level of CLPX was set to 100% 24 h post transfection in control. Error bars represent the standard error of the mean (SEM), $n = 3$. **b** In vitro degradation of CLPX by $LONM_6$ protease (400 nM) in the absence or presence of 1 μM PDIP38. Samples were separated by 10% Tris-Tricine SDS-PAGE and analysed by CBB staining. *LONM impurity. **c** Quantitation of in vitro degradation of CLPX by $LONM_6$ (400 nM) in the absence (open symbols, blue crosses represent individual data points) or presence (closed symbols, red crosses represent individual data points) of untagged PDIP38. Samples were separated by SDS-PAGE and analysed by CBB staining. Error bars represent the SEM of three independent experiments.

the recovery of untagged PDIP38 to either immobilised $ZBD_{4A}$ (Fig. 3f, lane 3) or $CLPX_{4A}$ (Fig. 3f, lane 7) was completely abolished, when compared to wild-type ZBD (Fig. 3f, lane 1) or CLPX (Fig. 3f, lane 5). Collectively, these data suggest that this region, which we refer to as the "adaptor-docking loop" of CLPX, performs a conserved function in bacterial and eukaryotic homologues of CLPX. Specifically, the ZBD of human CLPX forms a crucial docking platform for interaction with the putative adaptor protein PDIP38. To ensure that this change to PDIP38 docking was not due to overall dysfunction of the ZBD or to a dramatic change in its substrate specificity, we examined the peptide binding specificity of wild-type and mutant ZBD using a cellulose-bound (13-mer) peptide library (Fig. 3g, Supplementary Fig. 4 and Supplementary Fig. 5). Similar to most chaperones and proteases, the ZBD favoured recognition of peptides bearing a short (2–4 residues long) hydrophobic core (Fig. 3g and Supplementary Fig. 4); however, in contrast to most chaperones[62], the ZBD favoured peptides with an overall negative charge (Fig. 3g, compare spots 30 and 31). Importantly, the peptide binding specificity of both wild-type ZBD (Fig. 3g, upper panel) and $ZBD_{4A}$ (Fig. 3g, lower panel) was essentially identical, demonstrating that mutation of the adaptor-docking loop specifically inhibited PDIP38 docking without affecting peptide binding. Collectively, these data suggest that PDIP38 docks to CLPX, likely as an adaptor protein for the CLPXP protease.

**PDIP38 inhibits the LONM-mediated turnover of CLPX in vitro and stabilises the steady-state levels of CLPX in cells.** Next, we asked the question what is the consequence of the

interaction (between PDIP38 and CLPXP) in cells? To address this question, we knocked down PDIP38 expression in human (HeLa) cells using small interfering RNA (siRNA). Following successful knock down of PDIP38 (Fig. 4a, middle panel, compare lanes 1–3 with lanes 4–6), using the PDIP38-specific siRNA (#22994, Thermo Fisher Scientific), we analysed the steady-state levels of selected mitochondrial proteins. From this analysis, we identified that the levels of CLPX were reduced in HeLa cells transfected with the PDIP38-specific siRNA (Fig. 4a, upper panel, lanes 1–3) when compared to the levels of CLPX in HeLa cells transfected with a control siRNA (Fig. 4a, lanes 4–6). Importantly, this change was specific to CLPX as the levels of CLPP (Fig. 4a, lower panel) and the cross-reactive band recognised by the anti-PDIP38 antisera (Fig. 4a, middle panel, *) were unchanged by PDIP38 knockdown. These data suggest that PDIP38 plays an important role in maintaining the steady-state levels of CLPX in the cell. To validate this finding, we compared the steady-state levels of CLPX in cells treated with an alternative PDIP38-specific siRNA (either s25055 or s25056) with an additional control siRNA (Supplementary Fig. 6). Significantly, the loss of CLPX (as a result of PDIP38 knockdown) was specific, as the steady-state levels of two unrelated proteins (i.e. mitochondrial SDHA and the cytosolic protein, GAPDH) were not affected (Supplementary Fig. 6b, lower panels). Interestingly, these data are reminiscent of an unrelated bacterial adaptor protein—ClpS, which protects its cognate unfoldase (ClpA) from autocatalytic degradation in vivo[52]. Therefore, in order to determine if the levels of CLPX were regulated by its autocatalytic turnover, we examined the stability of CLPX in vitro, in the presence of CLPP with or without the addition of PDIP38 (Supplementary Fig. 7a).

However, in contrast to the idea that CLPX is degraded auto-catalytically, the in vitro stability of CLPX was unchanged by the presence of CLPP (Supplementary Fig. 7a). Next, in light of recent findings[63], which suggested that CLPX is a substrate of LONM, we speculated that PDIP38 might regulate the steady-state levels of CLPX in vivo by inhibiting its LONM-mediated turnover. To explore this possibility, we initially monitored the in vitro stability of CLPX in the presence of LONM, with or without the addition of PDIP38 (Fig. 4b). Consistent with the recent findings[63], CLPX was degraded by LONM with a half-life of ~60 min (Fig. 4c, open circles). Crucially, the LONM-mediated turnover of CLPX was inhibited by the addition of PDIP38 (Fig. 4b, lanes 8–13; Fig. 4c, filled circles). Moreover, the PDIP38-mediated inhibition of LONM was specific to CLPX turnover, as the addition of PDIP38 had no effect on the LONM-mediated turnover of casein (Supplementary Fig. 7b). Finally, we compared the steady-state levels of CLPX in cells lacking PDIP38, either in the presence of normal or depleted levels of LONM (Supplementary Fig. 6c). Consistent with our in vitro findings, depletion of LONM (by siRNA knockdown) partially restored the steady-state levels of CLPX in cells lacking PDIP38 (Supplementary Fig. 6c, compare lanes 2 and 3). Therefore, the PDIP38-mediated inhibition of CLPX turnover is in part due to PDIP38 shielding an exposed degron within CLPX that is normally recognised by LONM. Given the location of PDIP38 docking, this suggests that the CLPX degron is most likely located within the ZBD of CLPX. Collectively, these data suggest that the cellular levels and activity of mitochondrial CLPX (P) are not only regulated by the activity of LONM, but also by the levels of mitochondrial PDIP38.

**The N-terminal domain (NTD) of PDIP38 is essential for interaction with CLPX.** Next, to better understand the molecular basis of the interaction between CLPX and PDIP38, we examined the domain structure of PDIP38 and determined which domain (or domains) is required for docking to the ZBD of CLPX. To identify domain boundaries of PDIP38, we performed limited proteolysis (using thermolysin) of the mature protein (Supplementary Fig. 8). This approach demonstrated that PDIP38 is composed of two stable structural domains, which, based on the transient appearance of two intermediate fragments ($f_1'$ and $f_2'$) are likely to be joined by an exposed linker. To identify the boundary of these two domains, we performed six rounds of Edman degradation on the $f_1$ fragment, revealing the sequence FLANHD. Based on this analysis, we defined fragment $f_1$ as the C-terminal DUF525 domain and generated two GST-fusion proteins (Fig. 5a), the NTD of PDIP38 (residues 52–153) fused to the C terminus of glutathione S-transferase (GST) (GST-PDIP38$_N$) and the C-terminal domain of PDIP38 (residues 157–368) fused to the C terminus of GST (GST-PDIP38$_C$). To determine which domain was required for docking to CLPX, we performed a series of pull-down assays, in which H$_{10}$CLPX was immobilised to Ni-NTA agarose beads and then incubated with a bacterial cell lysate containing either overexpressed GST-PDIP38, GST-PDIP38$_N$ or GST-PDIP38$_C$ (Fig. 5b, lanes 2, 4 and 6, respectively). As a control, the different GST-PDIP38 fusion proteins were also incubated with Ni-NTA agarose beads lacking immobilised protein (Fig. 5b, lanes 3, 5 and 7, respectively). As expected, and consistent with Fig. 2a, full-length GST-PDIP38 was specifically eluted from the column containing immobilised H$_{10}$CLPX (Fig. 5b, lane 2). Importantly, deletion of the NTD of PDIP38 (GST-PDIP38$_C$) was sufficient to prevent any specific interaction between the two proteins (Fig. 5b, compare lanes 6 and 7). Consistent with these results, the NTD of PDIP38 alone was sufficient for interaction with CLPX as GST-PDIP38$_N$ was specifically recovered from Ni-NTA agarose beads containing

immobilised CLPX (Fig. 5b, lane 4) and not from beads lacking immobilised protein (Fig. 5b, lane 5). Taken together, these data clearly demonstrate that the NTD of PDIP38 docks to the ZBD of CLPX. Next, in order to better understand PDIP38 function, we crystalised mature PDIP38 (residues 52–368) and solved its structure by X-ray crystallography to 3.4 Å resolution (see Table 1 for statistics). Consistent with our biochemical analysis, the structure of PDIP38 is composed of two domains, an N-terminal YccV-like domain (residues 64–186) and a C-terminal DUF525 domain (residues 231–368), which are separated by a short linker region (Fig. 5c). The N-terminal YccV-like domain forms an antiparallel β-sheet structure composed of six β-strands (β0–β5–β1–β2–β3–β4), in which strands β0 to β4 are connected by loops and β4 and β5 is connected by a short 3$_{10}$ helix (Fig. 5c and Supplementary Fig. 9). Interestingly, in contrast to bacterial YccV (HspQ) homologues, the YccV-like domain of PDIP38 contains a large insertion between β2 and β3, which forms an extended structural element that contacts the C-terminal DUF525 domain. Specifically, β2/β3 extension forms an antiparallel β-sheet with β8 and the proximal sheet of the C-terminal DUF525 domain. Not surprisingly, the β8-strand of PDIP38 is also absent from bacterial ApaG homologues, suggesting that both of these regions have evolved in order to stabilise the interaction of the two PDIP38 domains. In addition to the extended β2/β3-sheet, the NTD of PDIP38 orthologues also contain an additional insertion, located between β3 and β4 (residues 143–166). This insertion is not only exposed in human PDIP38 (as it was susceptible to partial proteolysis), but is also highly flexible as it was not visible in the structure, presumably due to disorder. Based on the expected location of this loop, suspended over the putative substrate-binding pocket (see later) of the C-terminal DUF525 domain, we speculate that the L4 loop could play a role in regulating substrate-binding to the CTD. The linker (or intermediate) region, which connects the N- and C-terminal domains, is formed by a small N-terminal α-helix (α1), a two-stranded antiparallel β-sheet (β6 and β7) and a C-terminal α-helix (α2). This domain makes extensive contact to the NTD, wrapping around the domain, which appears to form a hinge point for movement of the CTD and potential delivery of its "cargo" to the associated ATPase component, CLPX. The CTD of PDIP38 (residues 231–368) forms an immunoglobulin-like fold. However, in contrast to its bacterial homologues (i.e. ApaG), which form a seven-stranded β-sandwich, PDIP38 contains an additional strand (β8), forming an eight-stranded β-sandwich composed of two four-stranded antiparallel β-sheets. The proximal sheet is composed of β8–β9–β10–β13, while the distal sheet is composed of strands β12–β11–β14–β15. Interestingly, although this domain exhibits only weak amino acid identity (~30%) with bacterial ApaG proteins and select eukaryotic F-box proteins with "other" domains (i.e. Fbxo3), all of these proteins share considerable structural homology (root-mean-squared deviation of ~1.5 Å for superposition of Cα atoms). Notably, the DUF525 domain of human Fbxo3 is essential for substrate recognition by the SCF-Fbxo3 ubiquitin ligase[64,65]. Therefore, to gain a better under-standing of PDIP38 function, we compared the known structure of human Fbxo3 DUF525 domain with our structure of human PDIP38 DUF525 domain and several other bacterial ApaG proteins (Supplementary Fig. 10). From this analysis, we identified a conserved hydrophobic groove, located between the two β-sheets of the C-terminal domain (Fig. 5d and Supplementary Fig. 10). Significantly, all but one of the nine hydrophobic residues are absolutely conserved from bacteria to humans (see Supplementary Table 1 and Supplementary Fig. 11), hence we propose that this groove plays an important role in substrate recognition. Consistently, the Fbxo3 substrate antagonist BC-1215 is proposed to dock into this conserved hydrophobic groove[66,67]. Based on its

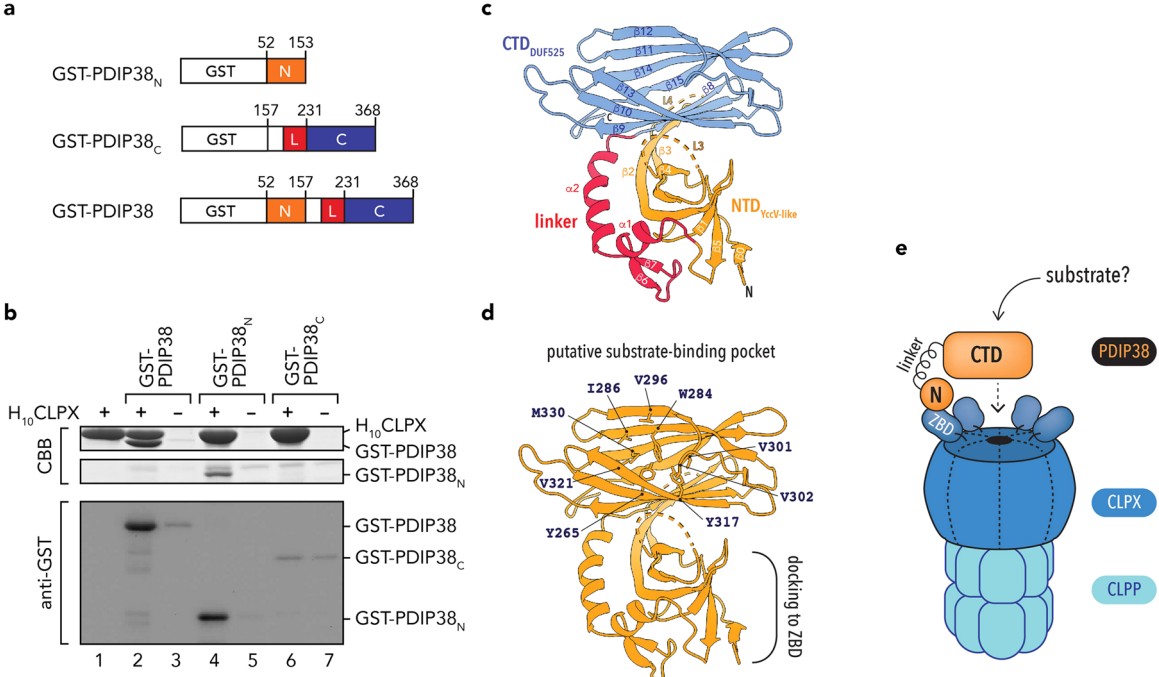

**Fig. 5 PDIP38 structure and proposed substrate delivery. a** Schematic representation of GST-PDIP38 fusion constructs (preprotein numbering is used). **b** In vitro pull-down using Ni-NTA agarose with (lanes 2, 4 and 6) or without (lanes 3, 5 and 7) purified immobilised H$_{10}$CLPX, incubated with *E. coli* lysate expressing GST-PDIP38 (lanes 2 and 3), GST-PDIP38$_N$ (lanes 4 and 5) or GST-PDIP38$_C$ (lanes 6 and 7). Eluted fractions are shown with samples analysed by Coomassie Brilliant Blue (CBB) staining or immunoblotting (with anti-GST) following separation by SDS-PAGE. As a control, purified H$_{10}$CLPX is shown in lane 1. Full gels and immunoblots are shown in Supplementary Fig. 17. **c** Ribbon representation of human PDIP38 highlighting its two domains. The N-terminal yccV-like (NTD$_{YccV-like}$) domain (orange) and the C-terminal DUF525 (CTD$_{DUF525}$) domain (blue) are separated by a hinge or linker region (red). Two unstructured loops (L3 and L4) within the NTD$_{YccV-like}$ domain are illustrated by dotted lines. **d** Ribbon representation of human PDIP38 (orange), highlighting the conserved hydrophobic residues (sidechains shown) that line the proposed substrate-binding groove of Fbxo3 (as described in Supplementary Table 1 and Supplementary Figs. 11 and 12). The docking site for interaction with the ZBD of CLPX is also indicated. **e** Cartoon representation of the interaction between mitochondrial PDIP38 and CLPX(P), illustrating the proposed recognition of a substrate via the CTD, for its delivery to CLPX, either for disassembly (via CLPX) or degradation (via CLPXP).

similarity to the substrate-binding domain of Fbxo3, we speculate that the CTD of PDIP38 is responsible for the recognition of as yet undefined proteins, and hence their delivery to CLPX, for removal by the CLPXP protease. Taken together, these data indicate that PDIP38 possesses key structural elements required to function as an adaptor of a AAA+ protein, specifically mitochondrial CLPX (Fig. 5e). PDIP38 contains a distinct NTD that docks to the accessory ZBD of CLPX (a well-established adaptor protein binding platform in bacterial homologues) and a distinct C-terminal domain, with likely capacity to support protein–protein interaction, to either, directly facilitate substrate delivery to CLPXP or indirectly regulate the CLPXP protease function.

## Discussion
We report the structural and functional characterisation of PDIP38, a component of the PN in mammalian mitochondria. Our biochemical analysis of PDIP38 shows that it is imported into the mitochondrion where is interacts specifically with the AAA+ unfoldase, CLPX (Fig. 1). PDIP38, however, is neither a substrate of CLPXP nor does it cause dissociation of the CLPXP complex (Fig. 2). We propose that PDIP38 is a modulator of the CLPXP protease. This proposal is based on our biochemical and cellular analysis of PDIP38, which demonstrate that PDIP38 not only fine-tunes the proteolytic activity of CLPXP in vitro, but also stabilises the cellular levels of CLPX by protecting CLPX from LONM-mediated degradation (Fig. 4). These features are commonly found in several bacterial AAA+ adaptor proteins

including the bacterial N-recognin, ClpS, which modulates both the specificity and the stability of its cognate unfoldase (ClpA) both in vitro and in vivo[30,52,68–70]. Our structural and biochemical studies demonstrate that PDIP38 is composed of two domains that are joined by a hinge region. The NTD is composed of a YccV-like fold, which is essential for interaction with CLPX. Interestingly, similar to the *E. coli* adaptor protein SspB[28,60,61], PDIP38 docks to a specific region within the ZBD of CLPX (termed the adaptor-docking loop). Mutation of this loop (in either the ZBD or full-length CLPX) abolishes PDIP38 docking, without affecting peptide binding by the ZBD (Fig. 3). Intriguingly, the YccV-like domain is found in several proteins that are involved in regulated proteolysis. In bacteria, YccV (also known as HspQ) was recently shown to be a substrate-like activator of the AAA+ protease, Lon[71]. Specifically, HspQ enhances the Lon-mediated turnover of specific substrates, including YmoA. However, in the absence of these substrates, HspQ is rapidly degraded by Lon[71]. Interestingly, under conditions of high acetyl-CoA, HspQ becomes acetylated and its turnover by Lon is inhibited[70]. This results in the accumulation of Ac-HspQ in the cell, which binds to ClpS and inhibits the turnover of N-degron substrates by ClpAP[70]. Remarkably, in *Arabidopsis thaliana* another YccV-like protein, termed ClpF is also believed to modulate the activity of ClpS[72], a putative plastid-localised N-recognin, for delivery of N-degron bearing substrates to the AAA+ protease, ClpCP[73,74]. Hence, it appears that YccV-like proteins not only regulate proteolytic pathways and/or machines in bacteria and the plastids of plants but also in human mitochondria as

**Table 1 Data collection and refinement statistics.**

|  | PDIP38 |
|---|---|
| Data collection |  |
| Space group | $P6_2$ |
| Cell dimensions |  |
| $a, b, c$ (Å) | 120.1, 120.1, 48.6 |
| $\alpha, \beta, \gamma$ (°) | 90, 90, 120 |
| Resolution (Å) | 39.31–3.39 (3.60–3.39) |
| $R_{sym}$ or $R_{merge}$ | 0.13 (1.44) |
| CC[a] in outermost shell | 73.1 |
| $<I/\sigma I>$ | 15.6 (2.39) |
| Completeness (%) | 99.8 (99) |
| Redundancy | 10.6 (10.6) |
| Refinement |  |
| Program | PHENIX |
| Resolution (Å) | 40–3.39 (4.27–3.39) |
| No. of reflections | 5690 |
| $R_{work}/R_{free}$ | 0.25/0.29 (0.29/0.33) |
| No. of atoms |  |
| Protein | 2001 |
| B-factors |  |
| Wilson (Å2) | 131 |
| Average $B$ factor (Å2): all atoms | 160 |
| R.m.s. deviations |  |
| Bond lengths (Å) | 0.002 |
| Bond angles (°) | 0.57 |
| Ramachandran statistics |  |
| Residues in favoured region, no. (%) | 92.8 |
| Residues in allowed region, no. (%) | 6.8 |
| Residues in outlier region, no. (%) | 0.4 |
| PDB entry | 6ZLX |

[a]Values within parentheses are for highest-resolution shell.

revealed by our data. Given the obvious evolutionary relationship of these proteolytic systems, it will be fascinating to see if PDIP38 plays any role in regulating a potential N-degron pathway in human mitochondria[48].

Similar to the NTD, the C-terminal domain of PDIP38 has also been identified in components of mammalian degradation pathways. In this case, the CTD is composed of an immunoglobulin fold (DUF525), which is not only found in bacterial ApaG proteins of unknown function but also occurs in a subset of Fbxo proteins (including Fbxo3). Significantly, Fbxo3 is the substrate recruitment component of a multidomain E3 ligase, termed the SCF complex (specifically the SCF-Fbxo3 complex). Importantly, Fbxo3 (and more specifically the DUF525 domain of Fbxo3) is essential for substrate (Fbxl2) recognition by the SCF-Fbxo3 complex[66,67]. Moreover, the recognition and ubiquitylation of Fbxl2 (by the SCF-Fbxo3 complex) can be completely abolished by docking of a small-molecule inhibitor (BC-1215) into the substrate-binding pocket of Fbxo3[66,67]. Importantly, the substrate pocket of Fbxo3 is conserved across all DUF525-containing proteins, including ApaG and mitochondrial PDIP38 (Supplementary Figs. 11 and 12). Hence, based on its similarity to Fbxo3, we speculate that the DUF525 domain of PDIP38 also facilitates the binding and delivery of a substrate to CLPXP (Fig. 5e). However, to date, we have yet to identify a ligand of PDIP38 that is delivered to CLPXP. Regardless of this speculative delivery function, PDIP38 clearly stabilises the cellular levels of CLPX (inhibiting its turnover by LONM) and modulates the substrate specificity of CLPXP in vitro. As such, we propose that mitochondrial PDIP38 is a regulator of the CLPXP protease. Interestingly, PDIP38 has recently been identified as a PrimPol-interacting protein; however, the relevance of this interaction in mitochondria is currently unclear, as the N-terminal mitochondrial targeting sequence of PDIP38 was identified (by chemical cross-linking experiments) as the major site of this interaction and the mature form of PDIP38 was unable to stimulate PrimPol DNA synthesis[75]. Therefore, many important questions about the function of mitochondrial PDIP38 still remain. What are the physiological substrates of mitochondrial PDIP38? In addition, PDIP38 is one of a growing number of mitochondrial proteins that also appears to "moonlight" in the nucleus[76]. The link, however, between the proposed function of PDIP38 within these two compartments remains a crucial question. How is the subcellular location of PDIP38 controlled and what is the significance of its dual localisation in cells? Similarly, given PDIP38 and Fbxo3 share a conserved domain (DUF525), which is essential for Fbxo3 function, this begs the question, is there crosstalk between these two proteins, outside the mitochondria? It will be fascinating to investigate the expression and targeting of PDIP38 to mitochondria (and the nucleus) in different cell types and/or at different developmental stages in mammals, not only in relation to the steady-state levels of CLPX, but also in relation to the turnover of Fbxo3 substrates. Finally, the future identification of physiological targets of the mitochondrial CLPXP/PDIP38 complex are eagerly awaited. These data will be invaluable to further develop our understanding of this system and its contribution to mitochondrial proteostasis.

## Methods

**Plasmids**. For in vitro transcription and translation of human PDIP38, pOTB7/*PDIP38* was obtained from the I.M.A.G.E. Consortium (ID 3349399). For the heterologous expression of PDIP38 in *E. coli*, the cDNA coding for mature PDIP38 (residues 52–368) was amplified by PCR from pOTB7/*PDIP38* using the appropriate primers (Supplementary Table 2) and cloned into either pHUE[49] between SacII and HindIII (to express untagged PDIP38), pET10N[77] between NotI and XhoI (to express PDIP38 with an N-terminal $H_{10}$ tag), pET10C[77] between NdeI and NotI (to express PDIP38 as a C-terminal $H_{10}$ fusion protein), pGEX-4T-1 between BamHI and XhoI (to express PDIP38 as an N-terminal GST-fusion protein) or pDD173[78] between NotI and HindIII (to express PDIP38 as a C-terminal GFP fusion protein with an N-terminal $H_{10}$ tag). To generate PDIP38$_N$ (residues 52–153) and PDIP38$_C$ (residues 157–368) fused to GST, pGEX-4T/*PDIP38* was subjected to site-directed mutagenesis[79] using primers PDIP_bam1 and PDIP_bam2 (see Supplementary Table 2). The resulting plasmid (pDT1367, see Supplementary Table 3) contained a stop codon and an additional BamHI site (and was used directly for the expression of GST-PDIP38$_N$). To generate GST-PDIP38$_C$, pDT1367 was digested with BamHI, the cut vector ligated lacking the *PDIP38$_N$* fragment to generate pDT1362. Plasmids for bacterial expression of human CLPX (full-length and domain mutants) and human CLPP (either His tagged and untagged) were described previously[80]. For expression of CLPX$_{4A}$ and ZBD$_{4A}$, pET10C/*hCLPX$_{4A}$* and pET10C/*hZBD$_{4A}$* were generated by site-directed mutagenesis using appropriate primers (see Supplementary Table 2). For details of primer sequences and plasmid constructs, refer to Supplementary information. All clones were confirmed by Sanger sequencing.

**Proteins**. Recombinant proteins were expressed, either in BL21-CodonPlus® (DE3)-RIL or XL1-Blue (Agilent) *E. coli* cells, grown in 2xYT media (containing appropriate antibiotic). Protein expression was induced with the addition of 0.5 mM isopropyl-β-D-thiogalactosidase at $OD_{600}$ ~0.8 and cultures were grown for at least 4 h at 20 °C. Following expression, His-tagged ($H_6$- or $H_{10}$-) recombinant proteins were purified from *E. coli* lysates under native conditions by immobilised metal affinity chromatography using Ni-NTA agarose (Qiagen) essentially as described[15] using 50 mM Tris-HCl [pH 8.0], 300 mM NaCl supplemented with an appropriate concentration of imidazole for binding (10 or 20 mM), washing (20 or 65 mM) and elution (250 or 500 mM). Purified $His_6$-Ub-PDIP38 and $His_6$-Ub-CLPP were cleaved using $His_6$-Usp2cc[49] and the untagged mature proteins recovered via a method outlined previously[49,69]. GST-PDIP38 was purified by affinity chromatography using GSH agarose (Bioserve) as outlined by the manufacturer. Radiolabelled PDIP38 preprotein was synthesised using TNT® SP6 Quick Coupled Transcription-Translation System (Promega) with undigested pOTB7/*PDIP38* as template and 11 μCi of [35S]Met/CysEXPRE35S35S protein labelling mix (specific activity of >1000 Ci/mmol) from Perkin Elmer. Protein assay (Bio-Rad) was used to determine protein concentrations using bovine serum albumin (Thermo Scientific) as a standard. Protein concentrations refer to the protomer, unless otherwise stated. FITC-casein, thermolysin, Prot. K and hen egg white lysozyme were purchased from Sigma-Aldrich, and DNase I was purchased from Gold Biotechnology. SeeBlue® Plus2 pre-stained and Mark12TM unstained protein standards were from Life Technologies.

**Electrophoresis and protein detection**. Proteins were separated using either glycine- or Tricine-buffered[81] sodium dodecyl sulfate-polyacrylamide gel electrophoresis (SDS-PAGE). Protein samples in 1× SDS-PAGE sample buffer (80 mM Tris-HCl [pH 6.8], 2% (w/v) SDS, 5% (v/v) glycerol, 100 mM dithiothreitol (DTT) and 0.02% (w/v) bromophenol blue) were heat treated at 95 °C for 5 min before separation. For visualisation of proteins, gels were stained with Coomassie Brilliant Blue R250 solution (CBB) or transferred to polyvinyldiflouride (PVDF) membrane using semi-dry method for immunoblotting. Primary antibodies: anti-PDIP38 (POLDIP2; Abcam ab109805), anti-PDIP38 (125/88; generated in rabbit using purified recombinant PDIP38-H$_{10}$ as antigen), affinity-purified anti-CLPX[15], anti-LONM[15], anti-TIM23 (BD Biosciences), anti-SDHA (Invitrogen), anti-GST (GE Healthcare), and anti-GAPDH (Life Technologies) were used at 1:1000 for Western blotting. Peroxidase-coupled secondary antibodies: anti-rabbit, anti-mouse and anti-goat IgG (Sigma-Aldrich) were used at 1:5000 for Western blotting. Antibody complexes were detected using enhanced chemiluminescence detection reagents (GE Healthcare) and digital images captured using GeneSnap (SynGene) or Image Lab$^{TM}$ software (Bio-Rad). FITC-casein was detected by in-gel fluorescence (excitation 488 nm and emission 526 nm), while radiolabelled proteins were detected by exposing dried gels to phosphor screens. Imaging was performed using a Typhoon$^{TM}$ Trio variable mode imager and analysed using the ImageQuant software (GE Healthcare).

**Limited proteolysis**. H$_{10}$PDIP38 (0.1 mg/ml) was subjected to limited proteolysis using thermolysin (0.01 mg/ml) at 30 °C in 50 mM Tris-HCl [pH 7.0], 150 mM NaCl and 5 mM CaCl$_2$. To terminate the reaction, samples were treated with 2 mM phenylmethanesulfonyl fluoride (PMSF) and heated at 95 °C in 1× SDS-PAGE sample buffer.

**Degradation assays**. The CLPXP-mediated degradation of FITC-casein was performed essentially as described[26]. Briefly, 0.4 μM CLPX$_6$P$_{14}$ was preincubated (at 30 °C for 5 min) in proteolysis buffer (50 mM Tris-HCl [pH 8.0], 100 mM KCl, 20 mM MgCl$_2$, 1 mM DTT, 0.02% (v/v) Triton X-100, 10% (v/v) glycerol) with FITC-casein (0.3 μM) in the absence or presence of 2.4 μM untagged PDIP38. To initiate degradation, 5 mM ATP was added and samples were incubated at 30 °C for the times indicated. Reactions were terminated by the addition of 1× sample buffer and the proteins denatured at 95 °C for 5 min.

**In vitro binding analysis**. The in vitro binding analysis was adapted from the method outlined in ref. [82]. *Escherichia coli* cells containing expressed GST-PDIP38, GST-PDIP38$_N$, GST-PDIP38$_C$ or untagged PDIP38 were resuspended (5 ml/g wet weight of cells) in binding buffer (20 mM HEPES-KOH [pH 7.5], 100 mM K(OAc), 10 mM Mg(OAc), 10% (v/v) glycerol, 65 mM imidazole) supplemented with 0.5% (v/v) Triton X-100, EDTA-free protease inhibitor cocktail (Roche), 2 mM PMSF, and DNase I (10 μg/ml), and then subjected to chemical lysis with lysozyme (0.2 mg/ml). Cell-free lysates or purified untagged PDIP38, as appropriate, were applied to Ni-NTA agarose beads either lacking or containing immobilised H$_{10}$-tagged CLPX, CLPX$_{ZBD}$, CLPX$_E$, *ec*ClpX$_{ZBD}$, CLPX$_{4A}$ or ZBD$_{4A}$ and incubated with end-over-end mixing at 4 °C for 30 min. The beads were then washed with five bed volumes (BVs) of binding buffer supplemented with 0.5% (v/v) Triton X-100, followed by 10 BV of wash buffer (binding buffer supplemented with 0.25% (v/v) Triton X-100). Bound proteins were eluted with elution buffer (50 mM Tris-HCl [pH 8.0], 300 mM NaCl, 500 mM imidazole). For binding assays containing full-length wild-type or mutant CLPX, all buffers were supplemented with 2 mM ATP and 10 mM β-mercaptoethanol.

**Cell culturing and treatment**. HeLa cells (a kind gift from Prof. N. Hoogenraad and validated by Cellbank Australia) were cultured in Dulbecco's modified Eagle's medium (Life Technologies) supplemented with 10% (v/v) foetal calf serum at 37 °C under an atmosphere of 5% (v/v) CO$_2$. Transfection of plasmid (10 μg) or 10–20 nM synthetic siRNA (Life Technologies) was performed using Lipofectamine® 2000 Transfection Reagent (Life Technologies) as per the manufacturer's instructions and cells grown for a further 24–72 h, as indicated. For interference of *PDIP38* mRNA, three independent synthetic siRNA (Life Technologies) were used: Silencer No. 22994 and Silencer Select Nos. s25055 (s55) and s25056 (s56). The corresponding Silencer Negative Control and Silencer Select Negative Controls No. 1 (nc1) and No. 2 (nc2) were used. For analysis, cells were detached by trypsin treatment (0.25% (w/v) trypsin, 1 mM EDTA; Invitrogen) and washed cell pellets lysed using TC extraction buffer (50 mM Tris-HCl [pH 7.5], 375 mM NaCl, 1 mM EDTA, 1% (v/v) Triton X-100) freshly supplemented with 2 mM PMSF. Soluble lysate was collected and used for analysis.

**Mitochondrial isolation and manipulation**. Crude mitochondria were isolated from HeLa cells as described[15,83]. In vitro import[84] was performed at 37 °C with [$^{35}$S]Met/Cys-labelled preprotein and isolated mitochondria resuspended in import buffer (20 mM HEPES-KOH [pH 7.4], 250 mM sucrose, 5 mM Mg(OAc), 80 mM K(OAc), freshly supplemented with 10 mM Na succinate, 1 mM DTT, 2% (w/v) fatty acid-free BSA, 5 mM ATP and 5 mM methionine. A mix of valinomycin (2 μM) and oligomycin (10 μM) was used to dissipate the membrane potential. Following import, mitochondria resuspended in SEM (250 mM sucrose, 1 mM

EDTA, 10 mM MOPS-KOH [pH 7.2]) were treated with ~40 μg/ml Prot. K for 15 min at 4 °C. Mitoplasts were formed in nine parts EM buffer (10 mM MOPS-KOH [pH 7.2], 1 mM EDTA) to one part SEM buffer at 4 °C for 20 min with gentle pipetting[84]. For protease treatment, mitochondria in SEM buffer, mitoplasts in EM buffer and lysed mitochondria in SEM buffer with 0.5% (v/v) Triton X-100 were incubated on ice with 50 μg/ml Prot. K for the times indicated. Prot. K was inhibited by the addition of 2 mM PMSF and proteins were immediately precipitated with trichloroacetic acid for analysis.

**Immunoprecipitation**. Mitochondrial lysate in IP buffer (50 mM Tris-HCl [pH 7.5], 100 mM KCl, 10 mM Mg(OAc), 5% (v/v) glycerol) containing 0.5% (v/v) Triton X-100, 10 mM ATP and 2 mM PMSF was mixed with PAS covalently attached to antibodies (anti-PDIP38 or anti-CLPX) by end-over-end rotation for 1 h at 4 °C. Beads were washed with 3 BV of IP buffer containing 0.25% (v/v) Triton X-100, 10 mM ATP and 2 mM PMSF and antibody bound protein eluted using 1 BV of 50 mM glycine [pH 2.5].

**Peptide library**. To examine the peptide binding specificity of human CLPX ZBD, peptide libraries (JPT Peptide Technologies) composed of 13-mer peptides (overlapping by 10 residues) derived from mtSSB and EFTu were immobilised to a cellulose membrane (see Supplementary Figs. 4 and 5, respectively, for peptide sequences) and panned with either ZBD or ZBD$_{4A}$, essentially as described previously[33] with minor modifications. Each peptide library was incubated either with ZBD or ZBD$_{4A}$ (2.5 μM) in MP2 buffer (15.7 mM Tris-HCl [pH 7.6], 100 mM KCl, 20 mM MgCl$_2$, 5% (w/v) sucrose, 0.05% (v/v) Tween-20) for 30 min with gentle shaking at room temperature. Following transfer to a PVDF membrane as described in ref. [33], the bound proteins were detected by immunodecoration using anti-human CLPX antisera (1:1000 dilution in 3% (w/v) skim milk powder/TBS + 0.05% (v/v) Tween-20 (TBS-T)), washed in 1× TBS-T before being incubated with anti-rabbit IgG conjugated with horse (Sigma-Aldrich; 1:5000 dilution in 3% (w/v) skim milk powder/TBS-T) and visualised using GelDoc$^{TM}$ XR+ (Bio-Rad) imaging system and images captured using QuantityOne (Bio-Rad). Figures were generated by overlaying 4–5 membrane images with adjusted transparency (100%, 50%, 33%, 25%, 20%) to ensure that all images had equal contribution to the overall result.

**Fluorescent detected analytical ultracentrifugation**. The interaction of PDIP38 (GFP-PDIP38) with CLPX$_{TRAP}$ and CLPX$_{TRAP}$P was analysed via FD-AUC, essentially as described by ref. [85]. GFP-PDIP38 (50 nM), CLPX$_{TRAP}$ (500 nM) and CLPP (2.8 μM) were prepared in 350 μl of XP-AUC buffer (50 mM Tris-HCl [pH 8.0], 100 mM KCl, 20 mM MgCl$_2$, 0.02% (v/v) Triton X-100, 10% (v/v) glycerol, 1 mM DTT, 5 mM ATP) and loaded into chilled aluminium cells fitted with a two-channel charcoal/epon centrepiece and sapphire windows (adjusted to 125 Psi) containing 50 μl of FC-43 heavy oil (3M, ID no. 98-0204-0101-8). Loading holes were sealed with thin plastic covers and screws, and cells were loaded into a prechilled An-50 Ti rotor (Beckman Coulter). The density and viscosity (poise) of the buffer was experimentally determined to be 1.0394 and 1.902 × 10$^{-2}$, respectively, in a DMA 4100 densitometer and Anton Paar AMVn automated micro viscometer (MEP instruments) fitted with a 1.6 mm capillary tube and 1.5 mm ball. Sedimentation velocity experiments were performed at 10 °C using an XL-A analytical ultracentrifuge (Beckman Coulter) retrofitted with an Aviv Biomedical fluorescence detector. Samples were centrifuged at 725 × *g* to optimise gain settings, and radial scans were collected at 72,500 × *g* continuously between 5.8 and 7.3 cm using 2 × 10$^{-4}$ cm increments, with fluorescence counts being measured at each radial position (five averages). Data were fitted to a continuous c(s) and c(m) model using SEDFIT (http://www.analyticalultracentrifugation.com).

**Crystallisation, X-ray diffraction and structure determination**. To investigate the structure of PDIP38, crystal screening was performed and crystals were obtained using 20% (w/v) PEG 8000, 100 mM HEPES, pH 7.5. Crystals were frozen using the crystallisation condition containing 15% (w/v) glycerol. Derivatives were prepared after the transfer of a native crystal into a drop solution containing the crystallisation buffer and a final concentration of Pt salts (1 mM Pt solution from the "Pt screens" purchased from Hampton Research). Crystals were incubated for 1 h in these drops, transferred into cryo solution (see above), flash frozen in liquid nitrogen and data collected at 100 K at the Swiss Light Source (Villigen, Switzerland; beamline PXII). Data were recorded on a PILATUS 6M detector (Dectris, Baden-Daettwil, Switzerland) and data reduction was performed using the program package XDS[86,87]. The structure of PDIP38 was solved to 3.4 Å by single anomalous dispersion techniques using one Pt derivative for phasing. The model was refined using PHENIX[88]. Most of the structure was unambiguously assigned in the electron density map, except for residues 52–62 at the N terminus and the loop regions (L3 between residues 108–126 and L4 between residues 144–167), due to poor density. Table 1 provides the statistics for the X-ray data collection and final refined model. Structural figures were generated using ChimeraX_Daily.

**Statistics and reproducibility**. For degradation assays, the reported means values and standard error of the mean was calculated from three independent experiments.

**Reporting summary**. Further information on research design is available in the Nature Research Reporting Summary linked to this article.

## Data availability

Data supporting the findings of this manuscript are available from the corresponding author upon reasonable request. A reporting summary for this article is available as a Supplementary Information file. Atomic coordinates for the PDIP38 structure have been deposited in wwPDB under accession code PDB 6ZLX.

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

## Acknowledgements

This work was supported by an Australian Research Council (ARC) Discovery Project (DP0770013) to D.A.D. and K.N.T., and ARC Future Fellowship to K.N.T. (FT0992033) and an ARC Australian Research Fellowship to D.A.D. (DP110103936). P.R.S. and H.Z. were supported by a La Trobe University Postgraduate Award, E.J.B. and B.R.L. were supported by Australian Postgraduate Awards and T.S. was supported by a Cooperative Research Centre postgraduate award. We thank Dr. Clemens Vonrhein from the Buster development group for his help with handling of the crystallographic data and M. Miasari for cloning of *PDIP38_N* and *PDIP38_C* into pGEX-4T.

## Author contributions

Conceptualisation, D.A.D. and K.N.T.; methodology, D.A.D., K.N.T., M.A.P. and K.Z.; investigation, P.R.S., E.J.B., H.Z., V.J.S., L.J.V., T.S., B.R.L., L.M.A. and K.Z.; writing—original draft, D.A.D., K.N.T. and K.Z.; writing—review and editing, D.A.D., K.N.T. and K.Z.; supervision, project administration and funding acquisition, D.A.D. and K.N.T.

## Competing interests

The authors declare no competing interests.
