## [Peer Review File · Communications Biology]

Reviewers' comments:

Reviewer #1 (Remarks to the Author):

The current manuscript reports on the import of the polymerase interacting protein PDIP38 into mitochondria. The biochemical characterization of PDIP38 combined with crystallographic analysis revealed that PDIP38 is composed of two domains: whereas the N-domain interacts within the ZBD of CLPX, the C-domain depicts an immunoglobulin-like fold (the role of which has not been further investigated). Eventually, Strack et al. conclude that PDIP38 regulates CLPXP activity through inhibition of the LONM-mediated turnover of CLPX.

Comments:

- 1) Title: The term 'novel adapter-like modulator' is confusing and meaningless.
- 2) Title: Using the abbreviation PDIP38 without further explanation will not attract a broad readership.
- 3) Abstract: First sentences of abstract and introduction read like copy & paste. The authors should remove redundancies.
- 4) Intro: Strack et al. describe in detail the possible functions of PDIP38 but state at the very end 'However, currently little is known about the structure or function of PDIP38, its mechanism of action or its role in mitochondrial proteostasis.' The authors should focus more on the topic.
- 5) Intro: The second paragraph describes in detail the ubiquitin proteasome system. This machinery is absent in mitochondria, thus, this part of introduction can be summarized in one sentence or even omitted. On the other hand, ClpXP is hardly described in its architecture and its functioning as a protease, but is the major topic of the current work. Furthermore, diseases are listed one after another without explaining the relationship to the ClpXP system.
- 6) Intro (97): ZBD of ClpX needs introduction as well as the C-terminal DUF525-domain.
- 7) Intro (100): A putative substrate binding domain is suggested without prove: 'We speculate that this groove is responsible for the specific recognition of short hydrophobic degrons'. Additional experimental data are requested.
- 8) Intro: The introduction closes with listing another set of proteins in bacteria, plants and humans that might have the 'YccV N-domain'. In my opinion, this description is unimportant to the reader and certainly does not allow any conclusion about the evolution of the structural motif in PDIP38.
- 9) Results: Strack et al. confirmed that PDIP38 is imported into mitochondria and can be immunoprecipitated with CLPXP as well as with the ZBD motif. Association constants should be determined and included in a possible revised version.
- 10) Results: It is questionable to what extent the detailed chapter discussing identification of the two domains in PDIP38 is necessary, since the authors later on report on its crystal structure, which depicts much more insights at molecular resolutions.
- 11) Results: To assign possible physiological functions of PDIP38, Strack et al. describe experiments but without major breakthroughs. In my opinion, the obtained findings do not permit the authors' conclusions to functionally classify PDIP38.
- 12) Results: The crystal structure of PDIP38 has been solved at 3.1 Å resolution. I am concerned that nowadays most reported crystal structures solely refer to $cc1/2$ and do not take into account other important values as well. The high resolution shell should have $I/I(\sigma)$ of at least 2.0 and an $R(\text{merge})$ below 0.6 in the resolution bin of 3.2 - 3.1 Å for the here reported structure. The values by Strack et al. are between 3.34 - 3.08 Å resolution, which is far too large and do not fulfill my criteria: $I/I(\sigma) = 0.59$, $R(\text{merge}) = 2.62$ with a data redundancy < 4.0 and a CC^* in the outermost shell of 0.122. Taken together, the crystal structure does not reflect the quality of measured reflections and therefore, does not correspond at least to my guidelines.
- 13) The description of the crystal structure does not allow conclusions about possible substrate binding grooves and functioning of PDIP38.
- 14) The entire results section more reads like an M&M section or experimental progress report.

Reviewer #2 (Remarks to the Author):

The manuscript by Strack et al reports the crystal structure of the Polymerase delta interacting protein of 38 kDa (PDIP38) and its proposed function as a potential ClpX adaptor protein. The authors confirm that PDIP38 is a mitochondrial protein and specifically interacts with CLPX in the mitochondrial matrix. This observation is consistent with previous work by Cheng et al J Biochem 2005, who demonstrated localization of PDIP38 to the mitochondrial matrix. The crystal structure of PDIP38 determined at 3.08-Å resolution shows a two-domain structure composed of an N-terminal YccV/SH3-like beta-barrel and a C-terminal Ig-like domain (DUF525), as predicted bioinformatically. The C-terminal domain is largely identical to the ApaG domain found in bacterial ApaG and human FBxo3 (Krzysiak et al FEBS J 2016), an F-box protein required for the interaction and degradation of FBx12. The authors propose that a hydrophobic groove in the ApaG domain may be involved for the recognition and delivery of proteins bearing specific hydrophobic degrons to the ClpXP protease. Furthermore, the authors demonstrate biochemically that the N-terminal SH3-like domain of PDIP38 interacts with the Zn-binding domain of CLPX similar to what has been reported for bacterial ClpX adaptor proteins. The 3D structure of a binary complex was not reported. Consistent with a role as a ClpX adaptor protein, interaction with PDIP38 stabilizes the steady state levels of CLPX in vivo and inhibits the LONM-mediated turnover of CLPX in vitro. The latter are new findings that merit publication.

General comment

The crystal structure of PDIP38 appears somewhat tagged-on. The paper could be improved by better integrating structure with function to help the logical flow. As is, the structure description seems repetitive with previous paragraphs and does not add to the research design. It also seems that the "Result", "Discussion" and "Conclusion" were combined into one single results section, which should be labelled accordingly.

Major comments

1. A role of PDIP38 as partner protein required for substrate delivery to CLPX is not fully supported. Does the hydrophobic groove overlap with the FBx12 binding site in FBxo3? Based on previous and present structure insight, the authors could generate PDIP38 point mutants predicted to be impaired in α S2casein binding resulting in α S2casein degradation (c.f. Fig 4a).
2. The N-terminal YccV-like domain of PDIP38 is structurally related to HspQ (Abe et al FEBS 2017), which was proposed to be a substrate for the Lon AAA protease (Puri and Karzai Mol Cell 2017). Can the authors exclude the possibility that PDIP38 is not a substrate for Lon and that CLPX binding may protect PDIP38 from Lon degradation?
3. Fig. 6: The figure legend does not match the figure shown. Key residues should be labelled in the surface representation. Are the structures shown in Fig 6c and 6d aligned? Fig 6e and f presumably only show the ApaG domain - what is their spatial relationship to Fig 6c and d? Presenting a stereo view would be helpful to provide an appreciation of the size and depth of the hydrophobic groove implicated in substrate binding.

Minor comments

1. ApaG is a conserved bacterial protein. The statement on line 106-107 that "the ApaG domain has been identified in Fbox-ONLY proteins such as Fbx3" is misleading. Furthermore, a reference is missing.
2. Line 169: Unless SEC-MALS was performed, SEC alone does not inform whether a protein is monodispersed.
3. Considering that the crystal structure of PDIP38 is presented here, the description of the bioinformatic approach (line 181-194) is superfluous. It is sufficient to state that the bioinformatic approach failed to correctly assign domain boundaries. Similarly, it is sufficient to state that siRNA

was unproductive in identifying specific PDIP38 interacting proteins (line 270-275).

4. There is a discrepancy of the resolution range reported in the paper (3.08Å) and in the validation report (3.34Å). Ramachandran outliers need to be corrected.
5. Line 333ff: how was sequence/structural homolog assessed and analyzed?
6. Line 343: PDB codes should be accompanied by the actual publication.
7. Line 347 and 348: please define "absolutely conserved". The term "conserved" is overly used throughout the manuscript.
8. Line 350-353: the relevance of this statement in context of this manuscript is unclear.
9. Fig. S5a: residues shown should be labeled.

Reviewer #3 (Remarks to the Author):

Strack et al. establish PDIP38 as a mitochondrially located interactor for ClpXP, which prevents LONP-mediated degradation of ClpX. Specifically, they show that the interaction is mediated by the N-terminal ZBD in ClpX and by the N-terminal YccV-like domain of PDIP38. Finally, the authors present a crystal structure of human mature PDIP38.

This is a well-written manuscript which links the proteinbiochemical characterisation of the direct interaction with a cell-biological evaluation of a functional role. The data are presented in well-structured figures, the biochemical experiments are of high quality and the literature is adequately cited. I particularly liked the authors frequent comparative description of the functions of the human and bacterial protein homologues to support their functional annotation. Collectively, this manuscript will help to shed light on the role of human ClpX whose function and substrates are, unlike those of its bacterial homologues, still largely unknown; and the reported mechanistic findings will surely be of interest to the community.

However, I have the following major points that should be experimentally addressed prior to publication.

- My main concern is that there are currently no data at all to support the notion that the C-terminal domain of PDIP38 contributes to substrate recruitment. This is entirely speculative at this point (including statements about the putative hinge region or the hydrophobic surface, which could also just facilitate a stable interaction), and as such the significance of this work is currently limited. This aspect should be experimentally strengthened before publication (and if unsuccessful, be featured less prominently in text & abstract). Can the authors look for potential substrates by an IP with just this domain and test a few candidates in vitro if feasible? Or can they test if the PDIP38 C-terminal domain binds to substrates of Fbx3? Or can they make a chimera between the N-terminal domain of PDIP38 and another ClpX-adaptor-substrate-binding domain to show that ClpXP, when bound to the YccV-domain, is capable of accepting folded substrates for degradation?
- What is the affinity of the interaction of PDIP38 and ClpX? This would help to assess its functional role and compare it to other ClpX adaptors. The authors should also mention protein concentration used for the SEC experiments.
- The "PDIP38 stabilises ClpX by blocking its LonM-mediated degradation" hypothesis requires strengthening both in cells and in vitro. Does siLonP rescue the siPDIP38 effect on ClpX? Does LonP in vitro cleave another substrate (but not ClpX) just fine in the presence of PDIP38?
- The structure in Figure 6 is currently rather descriptive and contributes little additional value. I recommend that the authors explore whether they can map the binding site of the interaction also on PDIP38 (as they did for the ZBD). This would allow the authors to link their structure to their biochemical data, and enable it to add to the functional insights.

- The authors should include all uncropped blots and gels in the SI with molecular weights shown.

In addition, there are the following minor points that warrant attention.

- Page 2, line 33: The word „bifunctional“ should be changed as currently no two separate functions are known for the two domains
- Page 2, line 40: In the cellular experiment, the author show that the „presence“ of PDIP38 stabilises the levels of ClpX, but the experiment does not allow a conclusion as to whether this is achieved by the interaction with ClpX or by other means.
- Page 2, line 43: Many proteins have multiple roles in different cellular compartments (e.g. in different cell states or cells), so I suggest authors to not downplay the nuclear functions of PDIP38 (unless they believe that these earlier nuclear findings indeed have no merit).
- Page 3, line 72: The number of human E3 ligases is typically quotes as around 600. Please change or add a reference for the 1000.
- Page 4, line 105: It is unclear to me what the authors mean with „act as an anti-adaptor of the N-recognin“
- Page 5, line 142: „in cells“ could be used instead of „in vivo“ which is usually reserved for living organisms
- Page 5, line 130: It seems that in reference 13 in 2005 it was already shown that PDIP38 is located in the mitochondrial matrix by experiments very similar to the ones shown in Figure 1b. The authors should cite this work in the occasions where the mitochondrial matrix location is stated. I appreciate the authors' achievement of replicating these experiments to strengthen the conclusion, but creating the impression that this is a novel finding should be avoided.
- Page 8, line 243: „demonstrating that PDIP38 does not trigger dissociation“ is a bit strong as a conclusion from a functional assay the absence of any data on the actually complex formation. In order to support the conclusion that PDIP38 changes ClpXP substrate processing, the authors should show that all casein substrates (and especially kappa-casein) are not degraded by free ClpP under similar assay conditions e.g. in the absence of ATP or the absence of ClpX, in a control experiment.
- Page 8, line 249: I have been unable to find a reference for the term „adaptor binding loop“. Please add one (and if this a newly introduced expression, consider introducing it after it is shown to be important for PDIP38-binding).
- Page 10, line 307: „Substrate recognition by PDIP38“: Consider rephrasing as neither PDIP38 has „substrates“ nor is it established that PDIP38 is involved in the substrate recruitment for ClpX.
- Page 10, line 314. Subscript for 10.
- Page 10, line 322. Please specify what is meant by „unique“ (for human protein, within paralogues, within orthologues, ...)
- Page 12, line 368. „substrate specificity“ refers to the artificial casein substrates and this should be added here as a qualifier.
- Page 20, line 653. Please use „two“ domains to be consistent with the main text.
- Page 21, line 685. Media and induction conditions should be added.
- Page 22, line 736: Has the lysate not been clarified by centrifugation?
- Page 23, line 781: How were cryopreservation and derivatisation carried out?
- Figure 1: Please define mPDIP38 and pPDIP38 in the caption,
- Figure 1c and d: I appreciate that the endogenous IP is difficult with the seemingly low affinity PDIP38 antibody. Can the authors specify whether the IP has been seen consistently across several replicates?
- Figure 2a: Please consider adding the other domains of ClpX.
- Figure 2d: What is the y-axis of the graphs (if UV absorbance, this should be stated clearly). Moreover, the MW of ZBD elutes earlier than the 25 kDa standard (suggesting a higher MW), but the text mentions an apparent MW of 24 kDa. The authors should state how the apparent MWs were calculated, mentioned the expected and found MWs, and double-check the assignment of the ZBD. If the authors have access to a light scattering machine, assessing the complex stoichiometry would be a plus, but is not required from my side.
- Figure 3c: Are the two CBB strips from the same gel?
- Figure 4d: What is shown underneath the ZBD overlay?

- Figure 5a: The authors should add a loading control (as they did in a similar experiment in Figure S2b)
- Figure 6c, e and f: A colour legend should be added.
- Supplementary Table 1: There are a few inconsistencies between the Table 1 in the manuscript and the wwPDB validation report that should be addressed (resolution: 3.08 vs. 3.34 Å, R_{free}: 0.28 vs. 0.30, B-factor for protein atoms: 28.2 vs. 171 Å²)

We thank each of the reviewers for their helpful feedback on our manuscript. A point-by-point response for each of their concerns (in italics) is itemised below. We believe the revised manuscript is significantly improved as a result of these suggestions and hope that the manuscript is now suitable for publication in *Communications Biology*.

Reviewer #1

1) Title: The term 'novel adapter-like modulator' is confusing and meaningless.

We have changed the title as requested by the reviewer.

2) Title: Using the abbreviation PDIP38 without further explanation will not attract a broad readership.

We have included the full name (and abbreviation) in the title as requested.

3) Abstract: First sentences of abstract and introduction read like copy & paste. The authors should remove redundancies.

We have removed redundancies from the abstract and introduction. Moreover, in order to address this point, together with point #4 and #5 (below) we have also re-written much of the introduction.

4) Intro: Strack et al. describe in detail the possible functions of PDIP38 but state at the very end 'However, currently little is known about the structure or function of PDIP38, its mechanism of action or its role in mitochondrial proteostasis.' The authors should focus more on the topic.

We have re-written the introduction, placing a greater emphasis on the role of PDIP38 and CLPXP in mitochondrial proteostasis.

5) Intro: The second paragraph describes in detail the ubiquitin proteasome system. This machinery is absent in mitochondria, thus, this part of introduction can be summarized in one sentence or even omitted. On the other hand, ClpXP is hardly described in its architecture and its functioning as a protease, but is the major topic of the current work. Furthermore, diseases are listed one after another without explaining the relationship to the ClpXP system.

We have re-written the introduction, placing a greater emphasis on the role of PDIP38 and CLPXP in mitochondrial proteostasis.

6) Intro (97): ZBD of ClpX needs introduction as well as the C-terminal DUF525-domain.

Both the ZBD and the C-terminal DUF525 domain have been introduced.

7) Intro (100): A putative substrate binding domain is suggested without prove: 'We speculate that this groove is responsible for the specific recognition of short hydrophobic degrons'. Additional experimental data are requested.

We agree that our speculation that PDIP38 recognises a short hydrophobic degron is not experimentally supported. Therefore, we have removed this speculation. However, we maintain that the hydrophobic groove within the DUF525 domain of PDIP38 is likely a substrate recognition site and believe that this idea is supported, by a combination of our structural data presented in this manuscript (which demonstrates that the residues within the hydrophobic groove are highly conserved across all DUF525 homologues, from bacteria to humans) and past published data that clearly show that the conserved groove (as defined here, is also present in the DUF525 homolog, Fbxo3) is essential for substrate interaction and ubiquitylation via the SCF-Fbxo3 complex. Related to the identification of a PDIP38 substrate, we think it is important to note, that although PDIP38 was first identified almost 20 years ago the physiological function (including its interacting proteins) and subcellular distribution of PDIP38 has been highly controversial. Much of these controversies have been exacerbated by a lack of thorough *in vitro* biochemical or structural analysis. This study represents the first detailed structural and biochemical/mechanistic analysis of PDIP38 and we hope that these data will provide unique insight into the function of mitochondrial PDIP38.

8) *Intro: The introduction closes with listing another set of proteins in bacteria, plants and humans that might have the 'YccV N-domain'. In my opinion, this description is unimportant to the reader and certainly does not allow any conclusion about the evolution of the structural motif in PDIP38.*

The reviewer is correct, the above description does not allow any conclusion about the evolution of this structural motif. However, we believe the occurrence of these homologues in plants and bacteria is important, therefore we have revised this description in relation to our findings, and moved it to the discussion.

9) *Results: Strack et al. confirmed that PDIP38 is imported into mitochondria and can be immunoprecipitated with CLPXP as well as with the ZBD motif. Association constants should be determined and included in a possible revised version.*

To determine an estimate of the binding affinity of PDIP38 for interaction with ClpX(ZBD) we have performed a series of interaction studies using immobilised ZBD. These data (new Supplementary Figure 3) and the estimated binding affinity ($K_d \sim 1.8 \mu\text{M}$) are included in the revised manuscript.

10) *Results: It is questionable to what extent the detailed chapter discussing identification of the two domains in PDIP38 is necessary, since the authors later on report on its crystal structure, which depicts much more insights at molecular resolutions.*

This information was retained to present the work in more a “historical” context. However, as highlighted by the reviewer, the discussion surrounding the initial identification of its domain organisation is superfluous, given the crystal structure of PDIP38 is also presented in the manuscript. Therefore, we have removed much of the text (and figures) surrounding the identification of PDIP38’s two domains. We believe these changes have significantly improved both the integration of data and the overall flow of the revised manuscript.

11) *Results: To assign possible physiological functions of PDIP38, Strack et al. describe experiments but without major breakthroughs. In my opinion, the obtained findings do not permit the authors' conclusions to functionally classify PDIP38.*

We respectfully disagree. We believe the data provided in this study is sufficient to classify PDIP38 as an adaptor protein. Although we are yet to identify the physiological substrate(s) of PDIP38, we provide strong evidence that PDIP38 alters the substrate specificity of CLPXP. Not only does PDIP38 inhibit the rapid turnover of α_{s2} -casein, whilst retaining the turnover of κ -casein, it also prevents the LONM-mediated degradation of CLPX. We also show, that the removal of PDIP38 from human cells, is responsible for the loss of its partner protein CLPX. Collectively, these findings demonstrate that PDIP38 is an important regulator of CLPX stability and CLPXP activity not only *in vitro*, but also in cells.

12) *Results: The crystal structure of PDIP38 has been solved at 3.1 Å resolution. I am concerned that nowadays most reported crystal structures solely refer to $cc1/2$ and do not take into account other important values as well. The high resolution shell should have $I/I(\sigma)$ of at least 2.0 and an $R(\text{merge})$ below 0.6 in the resolution bin of 3.2 - 3.1 Å for the here reported structure. The values by Strack et al. are between 3.34 – 3.08 Å resolution, which is far too large and do not fulfill my criteria: $I/I(\sigma) = 0.59$, $R(\text{merge}) = 2.62$ with a data redundancy < 4.0 and a CC^* in the outermost shell of 0.122. Taken together, the crystal structure does not reflect the quality of measured reflections and therefore, does not correspond at least to my guidelines.*

We have refined the structure and the updated statistics are included in Supplementary Table 1 of the revised manuscript.

13) *The description of the crystal structure does not allow conclusions about possible substrate binding grooves and functioning of PDIP38.*

Our description of a possible substrate binding site in PDIP38 is based (a) on the identification of highly conserved residues within the DUF525 domain proteins, Fbxo3 and ApaG, and (b) the reported binding site of a Fbxo3 substrate inhibitor (BC-1215) that prevents substrate recognition and ubiquitylation (via the SCF-Fbxo3 complex). Taken together, we believe these data are sufficient to speculate that substrate recognition by PDIP38 occurs via the C-terminal DUF525 domain.

14) *The entire results section more reads like an M&M section or experimental progress report.* We have revised much of the results section to improve the manuscript.

Reviewer #2:

The manuscript by Strack et al reports the crystal structure of the Polymerase delta interacting protein of 38 kDa (PDIP38) and its proposed function as a potential ClpX adaptor protein. The authors confirm that PDIP38 is a mitochondrial protein and specifically interacts with CLPX in the mitochondrial matrix. This observation is consistent with previous work by Cheng et al J Biochem 2005, who demonstrated localization of PDIP38 to the mitochondrial matrix. The crystal structure of PDIP38 determined at 3.08-Å resolution shows a two-domain structure composed of an N-terminal YccV/SH3-like beta-barrel and a C-terminal Ig-like domain (DUF525), as predicted bioinformatically. The C-terminal domain is largely identical to the ApaG domain found in bacterial ApaG and human FBxo3 (Krzysiak et al FEBS J 2016), an F-box protein required for the interaction and degradation of FBxl2. The authors propose that a hydrophobic groove in the ApaG domain may be involved for the recognition and delivery of proteins bearing specific hydrophobic degrons to the ClpXP protease. Furthermore, the authors demonstrate biochemically that the N-terminal SH3-like domain of PDIP38 interacts with the Zn-binding domain of CLPX similar to what has been reported for bacterial ClpX adaptor proteins. The 3D structure of a binary complex was not reported. Consistent with a role as a ClpX adaptor protein, interaction with PDIP38 stabilizes the steady state levels of CLPX in vivo and inhibits the LONM-mediated turnover of CLPX in vitro. The later are new findings that merit publication.

General comment

The crystal structure of PDIP38 appears somewhat tagged-on. The paper could be improved by better integrating structure with function to help the logical flow. As is, the structure description seems repetitive with previous paragraphs and does not add to the research design. It also seems that the "Result", "Discussion" and "Conclusion" were combined into one single results section, which should be labelled accordingly.

We thank the reviewer for their helpful suggestions. We have made significant changes to the manuscript, including the removal of repetitive comments and addition of a separate discussion section.

Major comments

1. *A role of PDIP38 as partner protein required for substrate delivery to CLPXP is not fully supported. Does the hydrophobic groove overlap with the FBxl2 binding site in FBxo3? Based on previous and present structure insight, the authors could generate PDIP38 point mutants predicted to be impaired in α S2casein binding resulting in α S2casein degradation (c.f. Fig 4a).*

It appears the reviewer has misunderstood the casein degradation experiment. Casein is a model CLPXP substrate, which is recognised by CLPX. Unfortunately, PDIP38 does not interact with α S2casein (or any form of casein) rather PDIP38 inhibits the CLPX-mediated recognition of α S2casein (likely via steric hinderance), while permitting the recognition (and turnover) of κ -casein (likely due to κ -casein binding to a site in CLPX that is unaffected by PDIP38 docking). This effect is very similar to that of the *E. coli* ClpA-adaptor protein, ClpS (which inhibits the recognition of GFP-SsrA and casein by ClpA and permits the recognition of aggregated-MDH (Dougan *et al.*, 2002, *Mol Cell* 9, 673 – 683), but is also required for the recognition and delivery of N-degron substrates (Erbse *et al.*, 2006; *Nature* 439: 753 – 756; Wang *et al.*, 2008, *Mol Cell* 32, 406 – 414; Schuenemann *et al.*, 2009, *EMBO reports* 10, 508 – 514) by reprogramming of ClpA (Rivera-Rivera *et al.*, 2014, *Proc. Natl. Acad. Sci.* 111, E3853 – E3859).

2. *The N-terminal YccV-like domain of PDIP38 is structurally related to HspQ (Abe et al FEBS 2017), which was proposed to be a substrate for the Lon AAA protease (Puri and Karzai Mol Cell 2017). Can the authors exclude the possibility that PDIP38 is not a substrate for Lon and that CLPX binding may protect PDIP38 from Lon degradation?*

Yes, we can exclude that mature PDIP38 is a substrate of mitochondrial LON. As shown in Supplementary Figure 7 (in the revised manuscript), we do not observe any turnover of mature PDIP38 *in vitro*, when incubated in the presence of LONM (and ATP). These data also clearly demonstrate that

PDIP38 does not inhibit the proteolytic activity of LONM, as casein turnover by LONM, is unaffected by the presence of PDIP38.

3. Fig. 6:

The figure legend does not match the figure shown. Key residues should be labelled in the surface representation.

We have corrected these errors.

Are the structures shown in Fig 6c and 6d aligned?

Yes

Fig 6e and f presumably only show the ApaG domain - what is their spatial relationship to Fig 6c and d?

Yes, both 6c and d (now Supplementary Figure 12 c and a, respectively) are shown in the same orientation as 6e and f (now Supplementary Figure 12 d and e, respectively).

Minor comments

1. ApaG is a conserved bacterial protein. The statement on line 106-107 that "the ApaG domain has been identified in Fbox-ONLY proteins such as Fbx3" is misleading. Furthermore, a reference is missing. This was an inadvertent error. Indeed, Fbxo3 is not an Fbox-only protein (although this term is almost exclusively used in the literature), it is a member of the Fbox "other" proteins (which is used to classify Fbox proteins that do not contain either a WD40 or LRR domain). We have corrected this statement and included an appropriate reference (Jin et al., 2004) in the revised manuscript.

2. Line 169: Unless SEC-MALS was performed, SEC alone does not inform whether a protein is monodispersed.

We did not use SEC-MALS, therefore we have removed this statement.

3. Considering that the crystal structure of PDIP38 is presented here, the description of the bioinformatic approach (line 181-194) is superfluous. It is sufficient to state that the bioinformatic approach failed to correctly assign domain boundaries. Similarly, it is sufficient to state that siRNA was unproductive in identifying specific PDIP38 interacting proteins (line 270-275).

We have revised the text as suggested.

4. There is a discrepancy of the resolution range reported in the paper (3.08Å) and in the validation report (3.34Å). Ramachandran outliers need to be corrected.

We have corrected this error, and refined the new structure using data to 3.39 Å resolution. The new statistics are presented in Supplementary Table 1 of the revised manuscript.

5. Line 333ff: how was sequence/structural homolog assessed and analyzed?

The root-mean-squared deviation (rmsd) between the structures was determined using superposition of C α atoms. We have clarified this in the revised manuscript.

6. Line 343: PDB codes should be accompanied by the actual publication.

We apologise for this oversight, we have added the publications (associated with each PDB code) to the revised manuscript. Due to substantial changes to the manuscript, this information now appears in the Supplementary information.

7. Line 347 and 348: please define "absolutely conserved". The term "conserved" is overly used throughout the manuscript.

"Absolutely conserved" was intended to indicate that the residues were identical across the species compared. We have changed the text to clarify the statement. We have also reduced our use of "conserved" throughout the manuscript, as requested.

8. Line 350-353: the relevance of this statement in context of this manuscript is unclear.

We changed the text surrounding this statement. We hope that the revised statement, in the context of this manuscript is now clear.

9. Fig. S5a: residues shown should be labeled.

We have labelled the residues as requested in original Fig S5a (now Supplementary Figure 11).

Reviewer #3:

Strack et al. establish PDIP38 as a mitochondrially located interactor for ClpXP, which prevents LONP-mediated degradation of ClpX. Specifically, they show that the interaction is mediated by the N-terminal ZBD in ClpX and by the N-terminal YccV-like domain of PDIP38. Finally, the authors present a crystal structure of human mature PDIP38.

This is a well-written manuscript which links the protein biochemical characterisation of the direct interaction with a cell-biological evaluation of a functional role. The data are presented in well-structured figures, the biochemical experiments are of high quality and the literature is adequately cited. I particularly liked the authors frequent comparative description of the functions of the human and bacterial protein homologues to support their functional annotation. Collectively, this manuscript will help to shed light on the role of human ClpX whose function and substrates are, unlike those of its bacterial homologues, still largely unknown; and the reported mechanistic findings will surely be of interest to the community.

However, I have the following major points that should be experimentally addressed prior to publication.

- My main concern is that there are currently no data at all to support the notion that the C-terminal domain of PDIP38 contributes to substrate recruitment. This is entirely speculative at this point (including statements about the putative hinge region or the hydrophobic surface, which could also just facilitate a stable interaction), and as such the significance of this work is currently limited. This aspect should be experimentally strengthened before publication (and if unsuccessful, be featured less prominently in text & abstract). Can the authors look for potential substrates by an IP with just this domain and test a few candidates in vitro if feasible? Or can they test if the PDIP38 C-terminal domain binds to substrates of Fbx3? Or can they make a chimera between the N-terminal domain of PDIP38 and another ClpX-adaptor-substrate-binding domain to show that ClpXP, when bound to the YccV-domain, is capable of accepting folded substrates for degradation?*

We agree that the idea that the C-terminal DUF525 of PDIP38 is involved in substrate recognition is largely speculative, however this speculation was developed from (a) our biochemical analysis of PDIP38's domain structure and its mode-of-interaction with CLPX, (b) the fact that PDIP38 is not a substrate of CLPXP, (c) our structural analysis of PDIP38 and the conserved nature of the residues that line this pocket (in all DUF525 homologues), and (d) published evidence which shows that this domain and the conserved pocket within the DUF525 homologue Fbxo3 is required for recognition and ubiquitylation (via the SCF-Fbxo3 complex) of its substrate Fbxl2.

We have attempted to isolate potential substrates by IP, however as is the case for most chaperones and proteases, the interaction with substrates is generally too weak to survive these types of experiments, without chemical crosslinking or development of a substrate "trap" mutant. Therefore, we believed this is beyond the scope of this current study and will likely require an alternatively approach, such as a detailed metabolomic analysis in PDIP38 KO cell lines, as was used to identify a mtClpX substrate in yeast (Kardon et al., 2015, Cell 161, 858–867).

Although substrates of Fbxo3 may interact with PDIP38, it is unlikely they will be delivered to CLPXP for degradation, as adaptor-mediated delivery to a cognate AAA+ protease generally involves a specific downstream recognition motif that is recognised by the AAA+ unfoldase. Currently however, very few human CLPX substrates have been identified and more importantly no substrate recognition motif for human CLPX has been determined. This problem is further exacerbated by the fact substrate recognition by human CLPX is clearly very different to bacterial ClpX (Martin *et al.*, 2008; *Mol Cell*, 29, 441 – 450) and hence a bacterial ClpX recognition motif cannot be substituted. As such, the delivery of a bacterial substrate to human CLPXP has not been successful. Therefore, it is currently impractical to engineer a

substrate that would be recognised by either Fbxo3 or a bacterial adaptor protein and then delivered to a non-cognate AAA+ unfoldase. For that reason, given the lack of direct evidence for substrate binding by PDIP38, we have, as suggested by the reviewer, featured this aspect of the manuscript less prominently in the text and abstract.

- *What is the affinity of the interaction of PDIP38 and ClpX? This would help to assess its functional role and compare it to other ClpX adaptors. The authors should also mention protein concentration used for the SEC experiments.*

We have determined the affinity of the interaction between PDIP38 and the ZBD of ClpX to be $\sim 1.8 \mu\text{M}$ and added this information to the revised manuscript (new Supplementary Figure 3). This value is similar to that of other AAA+ adaptor proteins, and has been discussed in the revised manuscript. The amount of protein used in SEC experiments has been added to the figure legend of Figure 3 (in revised manuscript).

- *The “PDIP38 stabilises ClpX by blocking its LonM-mediated degradation” hypothesis requires strengthening both in cells and in vitro. Does siLonP rescue the siPDIP38 effect on ClpX? Does LonP in vitro cleave another substrate (but not ClpX) just fine in the presence of PDIP38?*

We believe this hypothesis is already well supported by our data. We have already shown that knock down of PDIP38 in cells causes a loss of CLPX levels. To determine the cause of this loss, we examined the turnover of CLPX *in vitro*, using purified components. From these *in vitro* data we were able to confirm that LONM and not CLPXP is responsible for the degradation of CLPX. Importantly, and consistent with our data from mammalian cells, the turnover of CLPX by LONM was inhibited by PDIP38. Significantly, the inhibition of LONM-mediated substrate turnover, by PDIP38 was specific to CLPX as the LONM-mediated *in vitro* turnover of casein was not affected by PDIP38 (refer to Supplementary Figure 7). Nevertheless, to further strengthen this hypothesis we have examined the stability of CLPX in a PDIP38/LONM double knock down. Consistent with our *in vitro* data, knock down of LONM (in the absence of PDIP38) stabilised ClpX levels in cells (see new Supplementary Figure 6).

- *The structure in Figure 6 is currently rather descriptive and contributes little additional value. I recommend that the authors explore whether they can map the binding site of the interaction also on PDIP38 (as they did for the ZBD). This would allow the authors to link their structure to their biochemical data, and enable it to add to the functional insights.*

Although it would be interesting to further define the binding site on the NTD of PDIP38, we believe this is beyond the scope of our current study and we do not feel that this information would provide additional insight into the function of PDIP38.

- *The authors should include all uncropped blots and gels in the SI with molecular weights shown.*

We have added all uncropped blots to the supplementary information. In many cases, our western blots are performed using pre-cut PVDF membranes (to save antisera). In these cases, the full pre-cut membrane strip is provided to the supplementary information.

In addition, there are the following minor points that warrant attention.

- *Page 2, line 33: The word „bifunctional“ should be changed as currently no two separate functions are known for the two domains*

As requested, we have removed the term bifunctional from the revised manuscript.

- *Page 2, line 40: In the cellular experiment, the author show that the „presence“ of PDIP38 stabilises the levels of ClpX, but the experiment does not allow a conclusion as to whether this is achieved by the interaction with ClpX or by other means.*

We agree with the reviewer, that PDIP38 stabilises the cellular levels of ClpX. However, our conclusion that ClpX is stabilised by interaction with PDIP38 is not based solely on these cellular data. To verify that the stabilisation of ClpX by PDIP38 (in cells) is achieved through a physical interaction with PDIP38, we have performed *in vitro* experiments with purified components (ClpX, PDIP38 and LONM). From

these *in vitro* data we have been able to show that (a) ClpX is a substrate of LONM (consistent with the findings of Zurita Rendon & Shoubridge (2018)) and (b) the addition of PDIP38 (which forms a complex with ClpX) can inhibit the LONM-mediated degradation of ClpX. Importantly, this inhibition of LONM (by PDIP38), is specific to the substrate ClpX as PDIP38 is unable to inhibit the LONM-mediated turnover of casein (see Supplementary Figure 7)

• *Page 2, line 43: Many proteins have multiple roles in different cellular compartments (e.g. in different cell states or cells), so I suggest authors to not downplay the nuclear functions of PDIP38 (unless they believe that these earlier nuclear findings indeed have no merit).*

We didn't mean to suggest that the nuclear findings have no merit. We have changed the text in the revised manuscript to reflect this.

• *Page 3, line 72: The number of human E3 ligases is typically quotes as around 600. Please change or add a reference for the 1000.*

This part of the original manuscript has been removed from the revised manuscript (as suggested by reviewer #1).

• *Page 4, line 105: It is unclear to me what the authors mean with „act as an anti-adaptor of the N-recognin“*

An anti-adaptor of the N-recognin (i.e. ClpS) is a protein that inhibits/modulates the recognition of N-degron bearing protein substrates by ClpS. This part of the original manuscript has been removed upon revision of the manuscript.

• *Page 5, line 142: „in cells“ could be used instead of „in vivo“ which is usually reserved for living organisms*

We have removed all incorrect references to *in vivo* data throughout the manuscript and replaced with in cells, as suggested.

• *Page 5, line 130: It seems that in reference 13 in 2005 it was already shown that PDIP38 is located in the mitochondrial matrix by experiments very similar to the ones shown in Figure 1b. The authors should cite this work in the occasions where the mitochondrial matrix location is stated. I appreciate the authors' achievement of replicating these experiments to strengthen the conclusion, but creating the impression that this is a novel finding should be avoided.*

It was not our intention to create the impression that PDIP38 was not previously shown to be a mitochondrial protein. However, given there are numerous examples, where PDIP38 is shown to be directed to different cellular locations (and not the mitochondrion), we thought it important to clearly demonstrate that PDIP38 is imported into the mitochondrion in a membrane potential dependent manner. In fact, this is the first time the *in vitro* import of PDIP38 has been described. Nevertheless, we have added an additional citation to the work of Cheng *et al.*, 2005 to acknowledge their initial identification of PDIP38 as a mitochondrial protein.

• *Page 8, line 243: „demonstrating that PDIP38 does not trigger dissociation“ is a bit strong as a conclusion from a functional assay the absence of any data on the actually complex formation. In order to support the conclusion that PDIP38 changes ClpXP substrate processing, the authors should show that all casein substrates (and especially kappa-casein) are not degraded by free ClpP under similar assay conditions e.g. in the absence of ATP or the absence of ClpX, in a control experiment.*

These data have already been published by our lab (Lowth *et al.*, 2012). We have now included reference to these data in the revised manuscript. For the reviewers convenience, we have included the published supplementary figure below (not for publication with this manuscript).

• Page 8, line 249: I have been unable to find a reference for the term „adaptor binding loop“. Please add one (and if this a newly introduced expression, consider introducing it after it is shown to be important for PDIP38-binding).

As suggested, we have revised the text surrounding the description of the ‘adaptor binding loop’ to come after we have shown it to be important for PDIP38-binding.

• Page 10, line 307: „Substrate recognition by PDIP38“: Consider rephrasing as neither PDIP38 has „substrates“ nor is it established that PDIP38 is involved in the substrate recruitment for ClpX.

Given, we have yet to identify a PDIP38 “substrate” we have revised the statement as suggested by the referee and removed reference to “substrate recognition” by PDIP38.

• Page 10, line 314. Subscript for 10.

We have corrected this error in the revised manuscript.

• Page 10, line 322. Please specify what is meant by „unique“ (for human protein, within paralogues, within orthologues, ...)

This statement referred to PDIP38 orthologues, we have amended it accordingly in the revised manuscript.

• Page 12, line 368. „substrate specificity“ refers to the artificial casein substrates and this should be added here as a qualifier.

This part of the manuscript has been changed in the revised manuscript.

• Page 20, line 653. Please use „two“ domains to be consistent with the main text.

We have corrected the figure legend in the revised manuscript.

• Page 21, line 685. Media and induction conditions should be added.

The media and induction conditions have been added to the revised manuscript.

• Page 22, line 736: Has the lysate not been clarified by centrifugation?

Yes, in this case we used the term clarified lysates to represent lysates that were recovered following cell lysis and centrifugation to removal insoluble material.

• Page 23, line 781: How were cryopreservation and derivatisation carried out?

We have added this information to the revised manuscript.

• Figure 1: Please define mPDIP38 and pPDIP38 in the caption,

We have redefined mature PDIP38 (mPDIP38) as mt-PDIP38 and premature PDIP38 (pPDIP38) as pre-PDIP38 in the revised figure, to improve inherent understanding of the abbreviations.

• Figure 1c and d: I appreciate that the endogenous IP is difficult with the seemingly low affinity PDIP38 antibody. Can the authors specify whether the IP has been seen consistently across several replicates?

Yes, the IP has been repeated on at least 3 separate occasions (3 independent experiments). Although all 3 experiments, exhibit weak recovery of the partner protein, it is important to recognise that the interaction between PDIP38 and CLPX was confirmed by several additional experiments, both *in vitro* and in cells.

• *Figure 2a: Please consider adding the other domains of ClpX.*

We have added the mitochondrial targeting sequence (mts) and the AAA+ domain to the cartoon in the revised manuscript.

• *Figure 2d: What is the y-axis of the graphs (if UV absorbance, this should be stated clearly). Moreover, the MW of ZBD elutes earlier than the 25 kDa standard (suggesting a higher MW), but the text mentions an apparent MW of 24 kDa. The authors should state how the apparent MWs were calculated, mentioned the expected and found MWs, and double-check the assignment of the ZBD. If the authors have access to a light scattering machine, assessing the complex stoichiometry would be a plus, but is not required from my side.*

The y-axis represents quantitation of the protein bands (from SDS-PAGE) not UV absorbance. This method was used to enable quantitation of each component alone and in the complex whilst ensuring that each elution profile was quantitated by the same method. All MWs were estimated from a standard curve (prepared using protein standards for gel filtration; Aldolase (158 kDa), Albumin (67 kDa), Ovalbumin (43 kDa), Chymotrypsinogen A (25 kDa), Ribonuclease A (13.7 kDa)), the void volume of the column was determined from the elution of Blue dextran (2 MDa), Kav values were determined and plotted against the MW (log) and the best fit of these data points were represented by an equation. The MWs of each protein/protein complex was then estimated from the peak fraction using this equation. The R squared for the equation = 0.940 and hence some points sit above or below the line of best fit, which in turn can result in somewhat anomalous MW estimates (when simply comparing the elution volume of the protein standards). To ensure the MW estimates are accurate, we have replotted the standard curve and updated the estimated MW of each protein/complex. The updated MW are as follows (ZBD = ~ 29 kDa, PDIP38 = 45 kDa, ZBD+PDIP38 = ~60 kDa) and these values (together with their theoretical MWs, 12.1, 38.3 and 62.5 kDa, respectively) have been added to the revised manuscript.

• *Figure 3c: Are the two CBB strips from the same gel?*

Yes they are, the slabs were shown to simply highlight the important information within the gel and for space consideration. The full gels/immunoblots are included in Supplementary Figure 17.

• *Figure 4d: What is shown underneath the ZBD overlay?*

Figure 4d showed the structure of the ZBD dimer. However, as this figure adds nothing to the overall understanding of the manuscript we have removed it from the revised manuscript.

• *Figure 5a: The authors should add a loading control (as they did in a similar experiment in Figure S2b)*

In Figure 5a (now Figure 4a and Supplementary Figure 16 in the revised manuscript), CLPP serves as a loading control for the experiment. It shows that all lanes are equally loaded, as there is no change to the levels of CLPP. The loading is also verified by the non-specific immunoreactive band that is detected by the PDIP38 antisera (*).

• *Figure 6c, e and f: A colour legend should be added.*

We have added a colour legend for Figure 6c, e and f (see Supplementary Figure 12 c, d and e in the revised manuscript).

• *Supplementary Table 1: There are a few inconsistencies between the Table 1 in the manuscript and the wwPDB validation report that should be addressed (resolution: 3.08 vs. 3.34 Å, R_{free}: 0.28 vs. 0.30, B-factor for protein atoms: 28.2 vs. 171 Å²)*

We have refined the structure and updated the Table accordingly.

REVIEWERS' COMMENTS:

Reviewer #1 (Remarks to the Author):

The authors have invested great efforts to improve the manuscript.

Comments:

1-10) OK

11) Thanks for clarification

12) The refined structure now corresponds to my guidelines

13) I disagree. Data are still insufficient to speculate about substrate recognition. This issue should be addressed in the discussions section.

Publication is recommended after minor revision.

Reviewer #2 (Remarks to the Author):

I am satisfied with the authors' responses to my previous queries. The new manuscript is substantially improved and the PDIP38 structure is now much better integrated. I only have a few comments for the authors to address, which do not impede on publication of the manuscript in Communications Biology.

1. There seems to be several "breaks" in the protein main chain (Fig. 5c/d). To appreciate the structure, either a stereo-figure or connecting the missing region with a "dotted line" to follow the chain tracing is needed. I also do not see a segment of the electron density map to appreciate the quality of the experimental data.

2. Fig. 5d: The nature and structure of the putative substrate-binding pocket cannot be appreciated from Fig 5d that also appears to be identical to Fig. S10b-left. This needs to be corrected for clarity. The hydrophobic pocket is better shown in Fig. S12A or S12B - maybe exchange those figures.

3. I did not see a validation report for their revised structure deposition (6ZLX), which is now reported at 3.4-Å resolution.

4. Deriving mechanistic insights for human ClpX/ZBD binding preferences from the peptide array data shown in Fig. S4 and S5 need to be done cautiously. Especially Fig. S5 appears to be a non-native interaction with bacterial EFTu. Mammalian CLPXP substrates have been reported in the literature, which would appear to be better suited for such analysis (Szczepanowska et al EMBO J. 2016; Hofsetz et al MCP 2020). Hence, the comment on line 452 needs to be revised to PDIP38:CLPXP or similar, assuming of course that PDIP38 is a substrate mediator and not simply a regulator of CLPXP.

Reviewer #3 (Remarks to the Author):

The authors present an amended and much improved manuscript which through mainly biochemical and structural analysis characterises roles of PDIP38's interaction with the mitochondrially localised chaperone ClpX.

Even though the „smoking gun“ with regards to PDIP38 being a substrate adaptor for ClpX is still missing, I am strongly convinced that the manuscript in its current form warrants publication in Communications Biology for two main reasons: Firstly, because there is the new solid finding that

PDIP38 does modulate substrate specificity of ClpX, so that the „substrate adaptor“ role is merely an extension to this finding and a credible and forward-looking hypothesis which has been toned down to adequate levels to my taste. Secondly, because the in vitro evidence provided for the interaction of PDIP38 and ClpX is strong and does move the field forward (I agree with the author's response to point 7 of the first reviewer). I am sure that the publication of this data will trigger a comprehensive search for substrates delivered to ClpX by PDIP38 which with the author's data is now a viable (even though not easily testable) hypothesis.

The manuscript includes various improvements with new data of which I only highlight the additional interaction studies with immobilised ZBD and additional controls (incl. the double know down and the in vitro degradation control). Moreover, the revised form of the manuscript has been strengthened and improved in the writing. Lastly, the structure has been re-refined (which was needed, but I tend to disagree with the first reviewer's point 12 on I/sigI and R(merge) being >2.0 and < 0.6 , respectively which are outdated criteria, see <https://www.ncbi.nlm.nih.gov/pmc/articles/PMC3457925/>). It is my experience that with these 3.x A structures electron density can improve substantially if the scaling is done according to the CC* (or CC1/2) criteria.

We thank the reviewers for their feedback on the revised manuscript. A point-by-point response is itemised below. We hope that the revised manuscript is now suitable for publication in *Communications Biology*.

Reviewer #1 (Remarks to the Author):

The authors have invested great efforts to improve the manuscript.

Comments:

1-10) OK

11) Thanks for clarification

12) The refined structure now corresponds to my guidelines

13) I disagree. Data are still insufficient to speculate about substrate recognition. This issue should be addressed in the discussions section.

Publication is recommended after minor revision.

To address the reviewers concern, we have added a qualifying statement to the discussion regarding the speculation of substrate recognition by PDIP38.

Reviewer #2 (Remarks to the Author):

*I am satisfied with the authors' responses to my previous queries. The new manuscript is substantially improved and the PDIP38 structure is now much better integrated. I only have a few comments for the authors to address, which do not impede on publication of the manuscript in *Communications Biology*.*

1. There seems to be several "breaks" in the protein main chain (Fig. 5c/d). To appreciate the structure, either a stereo-figure or connecting the missing region with a "dotted line" to follow the chain tracing is needed. I also do not see a segment of the electron density map to appreciate the quality of the experimental data.

As suggested, we have added a dotted line to the revised Fig. 5 c/d and provided a new figure showing the electron density map to Supplementary Fig. 10.

2. Fig. 5d: The nature and structure of the putative substrate-binding pocket cannot be appreciated from Fig 5d that also appears to be identical to Fig. S10b-left. This needs to be corrected for clarity. The hydrophobic pocket is better shown in Fig. S12A or S12B - maybe exchange those figures.

As suggested, we have exchanged a revised version of Fig. 5d with Fig. S12a

3. I did not see a validation report for their revised structure deposition (6ZLX), which is now reported at 3.4-Å resolution.

Apologies if this was not included in the last submission. The validation report for 6ZLX is now included.

4. Deriving mechanistic insights for human ClpX/ZBD binding preferences from the peptide array data shown in Fig. S4 and S5 need to be done cautiously. Especially Fig. S5 appears to be a non-native interaction with bacterial EFTu. Mammalian CLPXP substrates have been reported in the literature, which would appear to be better suited for such analysis

(Szczepanowska et al EMBO J. 2016; Hofsetz et al MCP 2020). Hence, the comment on line 452 needs to be revised to PDIP38:CLPXP or similar, assuming of course that PDIP38 is a substrate mediator and not simply a regulator of CLPXP.

Although peptide binding preferences of CLPX/ZBD (in the context of a peptide library) are unlikely to differ for peptides derived from bacterial or human substrates, we agree that these data should be used cautiously as the identified motif was not tested beyond its identification by peptide library binding.

Line 452 of the manuscript is located in the acknowledgements! Based on the comment (above), we assume the reviewer is referring to the sentence on lines 444/445 “*Finally, the future identification of physiological targets of mitochondrial PDIP38 and CLPXP are eagerly awaited.*” As suggested by the reviewer, we have replaced “*PDIP38 and CLPXP*” in the above sentence with “*CLPXP/PDIP38 complex*”.

Reviewer #3 (Remarks to the Author):

The authors present an amended and much improved manuscript which through mainly biochemical and structural analysis characterises roles of PDIP38's interaction with the mitochondrially localised chaperone ClpX.

Even though the „smoking gun“ with regards to PDIP38 being a substrate adaptor for ClpX is still missing, I am strongly convinced that the manuscript in its current form warrants publication in Communications Biology for two main reasons: Firstly, because there is the new solid finding that PDIP38 does modulate substrate specificity of ClpX, so that the „substrate adaptor“ role is merely an extension to this finding and a credible and forward-looking hypothesis which has been toned down to adequate levels to my taste. Secondly, because the in vitro evidence provided for the interaction of PDIP38 and ClpX is strong and does move the field forward (I agree with the author's response to point 7 of the first reviewer). I am sure that the publication of this data will trigger a comprehensive search for substrates delivered to ClpX by PDIP38 which with the author's data is now a viable (even though not easily testable) hypothesis.

We thank the reviewer for their comments and their support for our speculation surrounding the putative hydrophobic groove in PDIP38. We think this support warrants inclusion of this speculation in the manuscript, with the aim to drive further research on this aspect of PDIP38, in the future.

The manuscript includes various improvements with new data of which I only highlight the additional interaction studies with immobilised ZBD and additional controls (incl. the double know down and the in vitro degradation control). Moreover, the revised form of the manuscript has been strengthened and improved in the writing. Lastly, the structure has been re-refined (which was needed, but I tend to disagree with the first reviewer's point 12 on I/sigI and R(merge) being >2.0 and < 0.6, respectively which are a outdated criteria, see <https://www.ncbi.nlm.nih.gov/pmc/articles/PMC3457925/>. It is my experience that with these 3.x A structures electron density can improve substantially if the scaling is done according to the CC (or CCI/2) criteria.*

We thank the reviewer for this comment and their support for our structural data.